# Approximate Secular Equations for the Cubic Regularization Subproblem

**Yihang Gao**
Department of Mathematics
The University of Hong Kong
Pokfulam, Hong Kong
gaoyh@connect.hku.hk

**Man-Chung Yue**
Musketeers Foundation Institute of Data Science
The University of Hong Kong
Pokfulam, Hong Kong
mcyue@hku.hk

**Michael K. Ng**
Department of Mathematics
The University of Hong Kong
Pokfulam, Hong Kong
mng@maths.hku.hk

## Abstract

The cubic regularization method (CR) is a popular algorithm for unconstrained non-convex optimization. At each iteration, CR solves a cubically regularized quadratic problem, called the cubic regularization subproblem (CRS). One way to solve the CRS relies on solving the secular equation, whose computational bottleneck lies in the computation of all eigenvalues of the Hessian matrix. In this paper, we propose and analyze a novel CRS solver based on an approximate secular equation, which requires only some of the Hessian eigenvalues and is therefore much more efficient. Two approximate secular equations (ASEs) are developed. For both ASEs, we first study the existence and uniqueness of their roots and then establish an upper bound on the gap between the root and that of the standard secular equation. Such an upper bound can in turn be used to bound the distance from the approximate CRS solution based ASEs to the true CRS solution, thus offering a theoretical guarantee for our CRS solver. A desirable feature of our CRS solver is that it requires only matrix-vector multiplication but not matrix inversion, which makes it particularly suitable for high-dimensional applications of unconstrained non-convex optimization, such as low-rank recovery and deep learning. Numerical experiments with synthetic and real data-sets are conducted to investigate the practical performance of the proposed CRS solver. Experimental results show that the proposed solver outperforms two state-of-the-art methods.

## 1 Introduction

The cubic regularization method (CR) is a variant of Newton's method proposed by Griewank [8], and later independently by Nesterov and Polyak [12], and Weiser et al. [16]. It gained significant attention over the last decade due to its attractive theoretical properties, such as convergence to second-order critical points[12] and quadratic convergence rate under mild assumptions [17]. Each iteration of CR solves a problem of the following form, called the cubic regularization subproblem (CRS):

$$\min_{\mathbf{x}\in\mathbb{R}^n} f_{\mathbf{A},\mathbf{b},\rho}(\mathbf{x}) := \mathbf{b}^{\mathrm{T}}\mathbf{x} + \frac{1}{2}\mathbf{x}^{\mathrm{T}}\mathbf{A}\mathbf{x} + \frac{\rho}{3}\|\mathbf{x}\|^3, \tag{1}$$

where $\rho > 0$ is the regularization parameter, $\mathbf{b} \in \mathbb{R}^n$, and $\mathbf{A} \in \mathbb{R}^{n \times n}$ is a symmetric matrix, not necessarily positive semidefinite. Many variants and generalizations of CR are developed, including

36th Conference on Neural Information Processing Systems (NeurIPS 2022).

the Adaptive Regularization Using Cubics (ARC) which allows for a dynamic choice of $\rho$ and inexact CRS solutions [3, 4], accelerated CR using momentum [15] and stochastic CR for solving stochastic optimization [14]. Despite the theoretical success, the practicality of CR and its variants relies critically on the CRS solver, a topic that attracts considerable research recently [2, 1, 10, 9]. The goal of this paper is to develop a novel, efficient CRS solver along with theoretical guarantees.

A popular approach for solving the CRS is via solving the so-called secular equation. We now review this approach. Towards that, we denote by $\lambda_1 \leq \cdots \leq \lambda_n$ the eigenvalues of $\mathbf{A}$ and by $\mathbf{v}_1, \cdots, \mathbf{v}_n$ the corresponding eigenvectors. In other words, we have the eigendecomposition $\mathbf{A} = \sum_{i=1}^{n} \lambda_i \mathbf{v}_i \mathbf{v}_i^T = \mathbf{V}\boldsymbol{\Lambda}\mathbf{V}^{\mathrm{T}}$, where $\boldsymbol{\Lambda} = \mathrm{diag}(\lambda_1, \ldots, \lambda_n)$ and $\mathbf{V} = [\mathbf{v}_1, \cdots, \mathbf{v}_n]$. Note that eigenvalues $\lambda_i$ are not necessarily positive due to the indefiniteness of the matrix $\mathbf{A}$. Also, we denote the Euclidean norm by $\|\cdot\|$.

**Proposition 1** ([12, 3]). *A vector $\mathbf{x}^*$ solves the CRS (1) if and only if it satisfies the system*

$$\begin{cases} (\mathbf{A} + \rho\|\mathbf{x}^*\|\mathbf{I})\mathbf{x}^* + \mathbf{b} = \mathbf{0}, & (2) \\ \qquad\qquad \mathbf{A} + \rho\|\mathbf{x}^*\|\mathbf{I} \succeq \mathbf{0}. & (3) \end{cases}$$

*Moreover, if $\mathbf{A} + \rho\|\mathbf{x}^*\|\mathbf{I} \succ \mathbf{0}$, then $\mathbf{x}^*$ is the unique solution (and hence a critical point).*

**Proposition 2** ([1]). *Let $\mathbf{x}^*$ be a global solution of CRS (1) and the eigendecomposition for $\mathbf{A} = \sum_{i=1}^{n} \lambda_i \mathbf{v}_i \mathbf{v}_i^T = \mathbf{V}\boldsymbol{\Lambda}\mathbf{V}^{\mathrm{T}}$, where $\boldsymbol{\Lambda} = diag(\lambda_1, \ldots, \lambda_n)$ and $\mathbf{V} = [\mathbf{v}_1, \cdots, \mathbf{v}_n]$. If $\mathbf{b}^{\mathrm{T}}\mathbf{v}_1 \neq 0$, then $\mathbf{A} + \rho\|\mathbf{x}^*\|\mathbf{I} \succ \mathbf{0}$ and the solution $\mathbf{x}^*$ is the unique critical point (and hence the unique solution). Conversely, if $\mathbf{b}^{\mathrm{T}}\mathbf{v}_1 = 0$, then the CRS (1) has multiple optimal solutions.*

From Proposition 2, if $\mathbf{b}^{\mathrm{T}}\mathbf{v}_1 \neq 0$, then there is only one critical point, which is also the optimal solution, and hence the gradient norm $\|\nabla f_{\mathbf{A}, \mathbf{b}, \rho}(\mathbf{x})\|$ serves as an optimality measure. Throughout the paper, we assume $\mathbf{b}^{\mathrm{T}}\mathbf{v}_1 \neq 0$, under which the CRS is said to be in the easy case. This is without much loss of generality as this holds generically true in practice. Moreover, we could easily avoid the hard case ($\mathbf{b}^{\mathrm{T}}\mathbf{v}_1 = 0$) by slightly perturbing the vector $\mathbf{b}$, see [12, 2].

To introduce the secular equation, note that in the easy case, conditions (2) and (3) can be written as

$$\begin{cases} (\boldsymbol{\Lambda} + \sigma\mathbf{I}) \cdot \mathbf{y}^* = \mathbf{c}, \\ \qquad\qquad \lambda_1 + \sigma > 0. \end{cases}$$

where $\sigma =: \rho\|\mathbf{x}^*\|$, $[y_1^*, \cdots, y_n^*]^{\mathrm{T}} := \mathbf{y}^* = \mathbf{V}^{\mathrm{T}}\mathbf{x}^*$ and $[c_1, \cdots, c_n]^{\mathrm{T}} := \mathbf{c} = -\mathbf{V}^{\mathrm{T}}\mathbf{b}$. Therefore,

$$y_i^* = \frac{c_i}{\lambda_i + \sigma}, \qquad i = 1, \ldots, n.$$

Since the Euclidean norm is invariant to orthogonal transformation, we have

$$\frac{\sigma^2}{\rho^2} = \|\mathbf{x}^*\|^2 = \|\mathbf{y}^*\|^2 = \sum_{i=1}^{n} \frac{c_i^2}{(\lambda_i + \sigma)^2}.$$

Consequently, instead of solving the complicated nonlinear system (2)-(3), we could solve the CRS (1) by first finding the (unique) root $\sigma > \max\{-\lambda_1, 0\}$ of the equation

$$w(\sigma) = \sum_{i=1}^{n} \frac{c_i^2}{(\lambda_i + \sigma)^2} - \frac{\sigma^2}{\rho^2}, \tag{4}$$

called the secular equation, and then solves the linear system $(\mathbf{A} + \sigma\mathbf{I})\mathbf{x} = -\mathbf{b}$. The first step can be done efficiently by using existing root-finding algorithms (e.g., the bisection method and Newton's method etc.).

The disadvantage of the above CRS solver, based on the secular equation (4), is that it requires the full spectrum of $\mathbf{A}$, which costs $\mathcal{O}(n^3)$. This approach is viable only for low- to moderate-dimensional problems. However, when $n$ is large, computing all eigenvalues of $\mathbf{A}$ is prohibitive. Worse still, after the root $\sigma$ is solved, we still need to apply iterative methods (e.g., Lanczos method) to solve the large-scale linear system (2). We are thus motivated to approximate the secular equation by using only some of the eigenvalues of $\mathbf{A}$, as opposed to all.

As our main contribution, we developed two different approximate secular equations (ASEs), both of which require computing $m < n$ eigenvalues of $\mathbf{A}$. The cost for forming the approximate secular

equations is only $\mathcal{O}(mn^2)$, and hence the resulting CRS solver is much more efficient and scalable. On the theoretical side, for each of the proposed approximate secular equations, we first studied the existence and uniqueness of its root, and then derived an upper bound on the gap between the root and that of the standard secular equation (4). This upper bound is in turn used to bound the distance from the approximate CRS solution based ASEs to the true CRS solution, thus offering a theoretical guarantee for the proposed CRS solver. A desirable feature of our CRS solver is that it requires only matrix-vector multiplication but not matrix inversion, which makes it particularly suitable for high-dimensional applications of unconstrained non-convex optimization, such as low-rank recovery and deep learning. On the empirical side, we conducted experiments with both synthetic and real problem instances to investigate the practical performance of the proposed CRS solver and the associated CR. Experimental results showed that the proposed solver outperforms two state-of-the-art methods. The selection of $m$ for the proposed ASEM is an interesting and crucial topic. We will discuss related issues in Section 4 and some numerical explorations are also presented in Section 5.

## 2 The First-Order Truncated Secular Equation

We define the first-order truncated secular equation by

$$w_1(\sigma; \mu) = \sum_{i=1}^{m} \frac{c_i^2}{(\lambda_i + \sigma)^2} + \sum_{i=m+1}^{n} \frac{c_i^2}{(\mu + \sigma)^2} - \frac{\sigma^2}{\rho^2}, \tag{5}$$

where $\mu \geq \lambda_m$ is an input parameter that approximates the unobserved eigenvalues $\lambda_{m+1}, \cdots, \lambda_n$, $c_i = -\mathbf{b}^{\mathrm{T}} \mathbf{v}_i$ and $\sum_{i=m+1}^{n} c_i^2 = \|\mathbf{b}\|^2 - \sum_{i=1}^{m} c_i^2$. Note that only $m$ eigenvalues $\lambda_1, \cdots, \lambda_m$ and their corresponding eigenvectors $\mathbf{v}_1, \cdots, \mathbf{v}_m$ are needed to form (5), which is computationally friendlier compared with (4). The name first-order truncated secular equation comes from the fact that $w_1(\sigma, \mu)$ is the first-order Taylor approximation to the function $w(\sigma)$. Below we will first study the existence and uniqueness for the root of (5). Then, we derive an error bound for the root.

### 2.1 Existence and Uniqueness for the Root

In the easy case that $\mathbf{b}^{\mathrm{T}} \mathbf{v}_1 \neq 0$ (equivalently, $c_1 \neq 0$), the solution $\mathbf{x}^*$ to the CRS (1) is unique, which implies the existence and uniqueness for the root $\sigma^*$ of (4). To show that our proposed CRS solver is also well-defined, we prove the existence and uniqueness of the root of the first-order truncated secular equation (5).

**Lemma 1.** *For any $\mu \geq \lambda_m$, the function $w_1(\cdot; \mu)$ as defined in (5) admits a unique root.*

*Proof.* **Existence.** We first consider the case when $\lambda_1 \leq 0$. Then, for any fixed $\mu \geq \lambda_m$,

$$\lim_{\sigma \to (-\lambda_1)^+} w_1(\sigma; \mu) = +\infty \quad \text{and} \quad \lim_{\sigma \to +\infty} w_1(\sigma; \mu) = -\infty,$$

By the intermediate value theorem, $w_1(\cdot; \mu)$ has a root in $(-\lambda_1, +\infty)$. For $\lambda_1 > 0$, we have

$$w_1(0; \mu) > 0 \quad \text{and} \quad \lim_{\sigma \to +\infty} w_1(\sigma; \mu) = -\infty.$$

Therefore, $w_1(\sigma; \mu)$ has a root in $(0, +\infty)$.

**Uniqueness.** Note that $w_1(\sigma; \mu)$ is monotonically decreasing for $\sigma \in (-\lambda_1, +\infty)$ and $\sigma \in (0, +\infty)$ when $\lambda_1 \leq 0$ and $\lambda_1 > 0$, respectively. Therefore, the uniqueness of the root for $w_1(\sigma; \mu)$ is guaranteed. $\square$

### 2.2 Error Analysis

In order to study the quality of the CRS solution based on our proposed solver using approximate secular equations, we need to study the quality of the root to the first-order truncated secular equation, denoted by $\sigma_1^*$. Towards that end, we provide an upper bound on the gap $|\sigma_1^* - \sigma^*|$ between $\sigma_1^*$ and the root $\sigma^*$ of the exact secular equation (4).

**Theorem 1.** *Let $\sigma_1^*$ and $\sigma^*$ be the unique roots of $w_1(\sigma; \mu)$ and $w(\sigma)$, respectively. Then*

$$|\sigma_1^* - \sigma^*| \leq C_m \cdot \max_{m+1 \leq i \leq n} |\lambda_i - \mu|, \tag{6}$$

where $C_m > 0$ is a constant, upper bounded by $\frac{2\|\mathbf{b}\|^2}{(\lambda_m - \lambda_1)^3} \cdot \min\left\{\frac{(\lambda_d + B_1)^3}{2\|\mathbf{b}\|^2}, \frac{\rho^2}{2B_1}\right\}$ with $B_1 = \frac{-\lambda_1 + \sqrt{\lambda_1^2 + 4\rho \cdot \|\mathbf{b}\|}}{2}$ being an upper bound for $|\sigma_1^*|$.

We clearly see that the right-hand side of inequality (1) is decreasing in $m$. This confirms that using more eigen information (i.e., larger $m$) helps to reduce the error $|\sigma_1^* - \sigma^*|$. The proof of Theorem 1 is technical and quite long and hence relegated to Appendix A. The approximation quality of our CRS solver is guaranteed by combining Theorem 1 with the following proposition.

**Proposition 3.** *Let $\mathbf{x}^*$ and $\tilde{\mathbf{x}}$ be solutions to the equations $(\mathbf{A} + \sigma^*\mathbf{I})\mathbf{x}^* = -\mathbf{b}$ and $(\mathbf{A} + \sigma_1^*\mathbf{I})\tilde{\mathbf{x}} = -\mathbf{b}$, respectively. Then, $\|\tilde{\mathbf{x}} - \mathbf{x}^*\| = \mathcal{O}(|\sigma_1^* - \sigma^*|)$.*

*Proof.* By definition, we have

$$\mathbf{x}^* = \sum_{i=1}^{n} (\lambda_i + \sigma^*)^{-1} \mathbf{v}_i \mathbf{v}_i^{\mathrm{T}} \cdot (-\mathbf{b}) = \sum_{i=1}^{n} (\lambda_i + \sigma^*)^{-1} c_i \cdot \mathbf{v}_i,$$

and

$$\tilde{\mathbf{x}} = \sum_{i=1}^{n} (\lambda_i + \sigma_1^*)^{-1} \mathbf{v}_i \mathbf{v}_i^{\mathrm{T}} \cdot (-\mathbf{b}) = \sum_{i=1}^{n} (\lambda_i + \sigma_1^*)^{-1} c_i \cdot \mathbf{v}_i,$$

then

$$\begin{aligned}
\|\tilde{\mathbf{x}} - \mathbf{x}^*\| &= \left\| \sum_{i=1}^{n} \left( (\lambda_i + \sigma_1^*)^{-1} - (\lambda_i + \sigma^*)^{-1} \right) \mathbf{v}_i \mathbf{v}_i^{\mathrm{T}} \cdot (-\mathbf{b}) \right\| \\
&\leq \left( \max_{1 \leq i \leq n} \left\{ (\lambda_i + \sigma_1^*)^{-1} - (\lambda_i + \sigma^*)^{-1} \right\} \right) \cdot \|\mathbf{b}\| \\
&= \mathcal{O}(|\sigma_1^* - \sigma^*|).
\end{aligned}$$

This completes the proof. $\qquad\square$

Before ending this section, some remarks are in order. First, the parameter $\mu$ acts as an approximation to $n - m$ unknown eigenvalues $\lambda_{m+1}, \cdots, \lambda_n$. An intuitive choice of $\mu$ that works well in practice and is computationally cheap is the average of unknown eigenvalues, i.e.,

$$\mu_1 = \frac{\sum_{i=m+1}^{n} \lambda_i}{n - m} = \frac{\mathrm{tr}(\mathbf{A}) - \sum_{i=1}^{m} \lambda_i}{n - m}. \tag{7}$$

Second, the error bound $C_m \cdot \max_{m+1 \leq i \leq n} |\lambda_i - \mu|$ in Theorem 1 depends on the distribution of eigenvalues of $\mathbf{A}$. If the unobserved eigenvalues $\lambda_{m+1}, \cdots, \lambda_n$ cluster around a small interval, then with a suitable choice of $\mu \in [\lambda_{m+1}, \lambda_n]$, $\max_{m+1 \leq i \leq n} |\lambda_i - \mu|$ is small. Conversely, if the unknown eigenvalues spread over a large interval, then it is hard to make the error $\max_{m+1 \leq i \leq n} |\lambda_i - \mu|$ small.

Third, it is instructive to study the error bound (6) under some random matrix model for $\mathbf{A}$. Suppose that $\mathbf{A} = \widetilde{\mathbf{A}}/\sqrt{2n}$, where $\widetilde{\mathbf{A}}$ is a symmetric random matrix with i.i.d. entries on and above the diagonal. By the Wigner semicircle law [6], as $n \to \infty$, the eigenvalues of $\mathbf{A}$ distribute according to a density of a semi-circle shape. In particular, we can deduce that with a probability of $1 - o(1)$,

$$\max_{m+1 \leq i \leq n} |\lambda_i - \mu| \leq \mathcal{O}\left( \left(1 - \frac{m+1}{n}\right)^{2/3} \right) \approx \left(\frac{3\pi}{4\sqrt{2}}\right)^{2/3} \cdot \left(1 - \frac{m+1}{n}\right)^{2/3} \tag{8}$$

The detailed proof of (8) and further discussions under random $\mathbf{A}$ can be found in Appendix C.

## 3  The Second-Order Truncated Secular Equation

Similarly to the equation (5), but with the second-order Taylor approximation, we define the second-order truncated secular equation by

$$w_2(\sigma; \mu) = \sum_{i=1}^{m} \frac{c_i^2}{(\lambda_i + \sigma)^2} + \sum_{i=m+1}^{n} \frac{c_i^2}{(\mu + \sigma)^2} - 2 \sum_{i=m+1}^{n} \frac{c_i^2 \cdot (\lambda_i - \mu)}{(\mu + \sigma)^3} - \frac{\sigma^2}{\rho^2}, \tag{9}$$

where $\mu \geq \lambda_m$ is an input parameter that approximates the unobserved eigenvalues $\lambda_{m+1}, \cdots, \lambda_n$.

### 3.1 Existence and Uniqueness for the Root

The lemma blew shows the existence and uniqueness of the root of $w_2(\cdot; \mu)$.

**Lemma 2.** *With*

$$\mu = \frac{\sum_{i=m+1}^{n} c_i^2 \cdot \lambda_i}{\sum_{i=m+1}^{n} c_i^2}, \tag{10}$$

*the function $w_2(\cdot; \mu)$ as defined in (9) admits a unique root.*

*Proof.* When

$$\mu = \frac{\sum_{i=m+1}^{n} c_i^2 \cdot \lambda_i}{\sum_{i=m+1}^{n} c_i^2},$$

the third summation in the definition (9) vanishes, and hence $w_2(\sigma, \mu)$ becomes the same as $w_1(\sigma, \mu)$, except with a specific choice of $\mu$. The desired conclusion then follows from Lemma 1. □

Unlike its first-order counterpart, we do not develop the existence and uniqueness of the root of the second-order truncated secular equation for arbitrary $\mu$. The reason is that when $\lambda_1 > 0$, $w_2(0; \mu)$ can potentially be positive or negative.

### 3.2 Error Analysis

Similar to that for the first-order truncated secular equation, we can also derive an error bound for the root of the second-order truncated secular equation.

**Theorem 2.** *Let $\sigma_2^*$ and $\sigma^*$ be the unique root of $w_2(\sigma; \mu)$ and $w(\sigma)$, respectively, and*

$$\mu = \frac{\sum_{i=m+1}^{n} c_i^2 \cdot \lambda_i}{\sum_{i=m+1}^{n} c_i^2}.$$

*Then,*

$$|\sigma_2^* - \sigma^*| \leq C_m \cdot \max_{m+1 \leq i \leq n} (\lambda_i - \mu)^2, \tag{11}$$

*where $C_m > 0$ is a constant bounded by $\frac{3\|\mathbf{b}\|^2}{(\lambda_m - \lambda_1)^4} \cdot \min\left\{\frac{(\lambda_n + B_1)^3}{2\|\mathbf{b}\|^2}, \frac{\rho^2}{2B_1}\right\}$ with $B_1 = \frac{-\lambda_1 + \sqrt{\lambda_1^2 + 4\rho \cdot \|\mathbf{b}\|}}{2}$ being an upper bound for $|\sigma_2^*|$.*

The proof of Theorem 2 can be found in Appendix B. We can similarly estimate the approximation quality by combining Theorem 2 and Proposition 3. Again, the right-hand side of the error bound (11) is decreasing in $m$. We should also point out that the CRS solver based on the second-order secular equation outperforms the first-order counterpart only if $\max_{m+1 \leq i \leq n} |\lambda_i - \mu|/|\lambda_m - \lambda_1|$ is small enough. The computation of $\mu$ here requires $c_{m+1}, \cdots, c_n$, which seem to be inaccessible. We provide a tractable form for $\mu$ in (13) and will discuss it in the next part.

## 4 Implementation Details

We now discuss the implementation details for solving the proposed first-order secular equation for CRS. First, we obtain the partial eigen information $\{\lambda_1, \cdots, \lambda_m\}$ and $\{\mathbf{v}_1, \cdots, \mathbf{v}_m\}$ by Krylov subspace methods. Note that only Hessian-vector products are required. This is computationally friendlier than other methods that rely on matrix inversions and is particularly suitable for modern, high-dimensional applications. Then, we solve the first-order secular equation (5) with $\mu$ defined in (7) or (10), using any root-finding algorithm, such as Newton's method. Finally, we solve the linear system $(\mathbf{A} + \sigma^* \mathbf{I})\mathbf{x} = -\mathbf{b}$ by iterative algorithms, e.g., the Lanczos method and the conjugate gradient method. The resulting CRS solver, namely the approximate secular equation method (ASEM), is summarized as follows:
**Step 1:** obtaining the partial eigen information $\{\lambda_1, \cdots, \lambda_m\}$ and $\{\mathbf{v}_1, \cdots, \mathbf{v}_m\}$ of $\mathbf{A}$.
**Step 2:** solving the secular equation (5) with $\mu$ defined in (7) or (10); we get $\sigma^*$.
**Step 3:** iteratively solving the linear system $(\mathbf{A} + \sigma^* \mathbf{I})\mathbf{x} + \mathbf{b} = \mathbf{0}$.
**Output:** the solution $\mathbf{x}$.

**Details for Step 1.** The Krylov subspace is one of the most popular iterative methods in solving eigen problems with $\mathcal{O}(mn^2)$ computational cost [13]. The Lanczos decomposition for a real symmetric matrix $\mathbf{B}$ satisfies

$$\mathbf{B}\mathbf{U}_k = \mathbf{T}_k\mathbf{U}_k + \beta_k\mathbf{u}_{k+1}\mathbf{e}_k^{\mathrm{T}},$$

where $\mathbf{U}_k \in \mathbb{R}^{n \times k}$ is an orthonormal matrix (i.e., $\mathbf{U}_k^{\mathrm{T}}\mathbf{U}_k = \mathbf{I}_k$), $\mathbf{T}_k \in \mathbb{R}^{k \times k}$ is a symmetric tridiagonal matrix and $\mathbf{e}_k \in \mathbb{R}^k$ is the $k$-th standard basis vector in $\mathbb{R}^k$. Lanczos observed that even for comparatively small $k$, $\mathbf{T}_k$ approximates $\mathbf{B}$ very well in terms of eigenvalues and eigenvectors. Specifically, for a suitable eigenpair $(\gamma, \mathbf{w})$ of $\mathbf{T}_k$ with $\mathbf{T}_k\mathbf{w} = \gamma \cdot \mathbf{w}$, the pair $(\gamma, \mathbf{U}_k\mathbf{w})$ is an approximate eigenpair of $\mathbf{B}$, i.e., $\mathbf{B}\mathbf{z} \approx \gamma \cdot \mathbf{z}$ with $\mathbf{z} = \mathbf{U}_k\mathbf{w}$. Here, the Krylov subspace is constructed by $\mathbf{u}_1$, the first column of $\mathbf{U}_k = [\mathbf{u}_1, \cdots, \mathbf{u}_k]$, i.e., $\mathcal{K}_k(\mathbf{B}, \mathbf{u}_1)$. Note that $\mathbf{T}_k$ approximates $\mathbf{B}$ for eigenvalues with largest modulus (or absolute values) and the corresponding eigenvectors. Empirically, for calculating $m$ eigenvalues of $\mathbf{B}$ with largest absolute values and the corresponding eigenvectors, we usually construct the Krylov subspace $\mathcal{K}_k(\mathbf{B}, \mathbf{u}_1)$ with dimension $k = \max\{2m, 20\}$. The base vector $\mathbf{u}_1$ is also essential for the Krylov subspace method. Moreover, restarting is adopted to iteratively update the base vector $\mathbf{u}_1$. To the best of our knowledge, the mentioned iterative method for partial eigen information is supported in many softwares, e.g., Matlab (eigs function) and Python (Scipy package) etc. For more details, please refer to ARPACK [11]. Returning back to the proposed algorithm with $\mathbf{A}$, instead of calculating the largest (in terms of the absolute value) $m$ eigenvalues of $\mathbf{A}$, we aim to get $m$ (algebraically) smallest eigenvalues $\{\lambda_1, \cdots, \lambda_m\}$ and the corresponding eigenvectors $\{\mathbf{v}_1, \cdots, \mathbf{v}_m\}$. We first roughly calculate a shift value $\beta \geq \|\mathbf{A}\|$ by several steps of power iteration (Hessian-vector products). Then, let $\mathbf{B} = \beta \cdot \mathbf{I} - \mathbf{A}$, whose eigenvalues $\{\beta - \lambda_n, \cdots, \beta - \lambda_1\}$ are non-negative and the corresponding eigenvectors are $\{\mathbf{v}_n, \cdots, \mathbf{v}_1\}$. Applying the mentioned Krylov subspace algorithm, we first obtain an estimate of shifted eigenvalues $\{\lambda_1, \cdots, \lambda_m\}$ and the corresponding eigenvectors $\{\mathbf{v}_1, \cdots, \mathbf{v}_m\}$ for $\mathbf{A}$, since $\{\beta - \lambda_m, \cdots, \beta - \lambda_1\}$ are largest eigenvalues of $\mathbf{B}$. To further lower the computational cost, we may adopt $k$-dimensional Krylov subspace $\mathcal{K}_k(\mathbf{B}, \mathbf{u}_1)$ for $m$ eigenvalues without restarting in implementation with $k = m$.

**Details for Step 2.** Instead of directly solving (5), Cartis et al. [3] recommended to find the root for the equivalent equation:

$$\tilde{w}_1(\sigma; \mu) = \sqrt{\sum_{i=1}^{m}\frac{c_i^2}{(\lambda_i + \sigma)^2} + \sum_{i=m+1}^{n}\frac{c_i^2}{(\mu + \sigma)^2}} - \frac{\sigma}{\rho}, \tag{12}$$

which is convex on $(-\lambda_1, +\infty)$. Moreover, under perfect initialization, Newton's method is proved to achieve (locally) quadratic convergence. However, we numerically find that it depends much on the initialization and may converge to a point outside the feasible domain $(-\lambda_1, +\infty)$ if it has imperfect initialization. Here, we recommend to use the bisection method to find the root of (5) or (12) due to its linear convergence, stability and ease of implementation. For the weighted average $\mu$ defined in (10), we can rewrite it as a more tractable but equivalent form:

$$\mu_2 = \frac{\sum_{i=m+1}^{n} c_i^2 \cdot \lambda_i}{\sum_{i=m+1}^{n} c_i^2} = \frac{\mathbf{b}^{\mathrm{T}}(\mathbf{A} - \mathbf{V}_m\boldsymbol{\Lambda}_m\mathbf{V}_m^{\mathrm{T}})\mathbf{b}}{\|\mathbf{b}\|^2 - \sum_{i=1}^{m} c_i^2} = \frac{\mathbf{b}^{\mathrm{T}}\mathbf{A}\mathbf{b} - \sum_{i=1}^{m} c_i^2 \cdot \lambda_i}{\|\mathbf{b}\|^2 - \sum_{i=1}^{m} c_i^2}, \tag{13}$$

where $\mathbf{V}_m = [\mathbf{v}_1, \cdots, \mathbf{v}_m]$ and $\boldsymbol{\Lambda}_{\mathbf{m}} = \mathrm{diag}(\lambda_1, \cdots, \lambda_m)$.

**Details for Step 3.** There are many well-studied, efficient and reliable iterative methods for (real symmetric) linear systems, e.g., Krylov subspace (Lanczos) methods and conjugate gradient methods etc. We adopt the Lanczos method for solving the linear system $(\mathbf{A} + \sigma^*\mathbf{I})\mathbf{x} + \mathbf{b} = \mathbf{0}$, where only a few steps of Hessian-vector products are required.

In summary, the main computational cost comes from Step 1 and Step 3 for Hessian-vector products $(\mathcal{O}(mn^2))$, since solving the root of $w_1(\sigma; \mu)$ is a 1-dimensional problem in Step 2 and is of cost $\mathcal{O}(n)$. Therefore, the total computational cost for the proposed algorithm is $\mathcal{O}(mn^2)$, much lower than the method based on full eigendecomposition $(\mathcal{O}(n^3))$.

**The selection of $m$.** The choice of the parameter $m$ is important to our CRS solver: a larger $m$ yields a better CRS solution quality but incurs a higher computational cost. If $\mathbf{A}$ is a Gaussian random matrix, by (8), we can achieve $\varepsilon$-accuracy (i.e., $|\sigma_1^* - \sigma^*| \leq \varepsilon$) if $m \leq n$ satisfies

$$\left(\frac{3\pi}{4\sqrt{2}}\right)^{2/3} \cdot \left(1 - \frac{m+1}{n}\right)^{2/3} \leq \varepsilon.$$

However, the error bound (8) provides only a conservative sufficient conditions for $m$. Moreover, for general problems without the Gaussian assumption on $\mathbf{A}$, it is hard to choose $m$ based on the $\varepsilon$-accuracy, because the error bounds (6) and (11) are implicit in $m$. Therefore, adaptive methods (or heuristic methods) for selecting $m$ are necessary in practice. A natural way is to check the suboptimality (gradient norm) in each step, and enlarge $m$ by $m_0$ (i.e., we set $m = \max\{m + m_0, m_{\max}\}$, where $m_{\max}$ is the maximal number of eigenvalues we adopt in ASEM), if the output does not satisfy the given condition for suboptimality. Moreover, numerical experiments on CUTEst problems (see Experiment 6 in Section 5) shows that $m = 1$ is enough for most of the cases. We left the study for the selection of $m$ as future work. To the best of our knowledge, the Krylov subspace method [3, 2] for CRS suffers from a similar issue of hyperparameter selection.

## 5 Experimental Results

Without the loss of generality (see Appendix D), we assume that $\mathbf{A}$ is diagonal in the synthetic CRS instances, for simplicity and fair comparison. Furthermore, we also test the proposed ASEM on CUTEst library [7]. All experiments were run on a Macbook Pro M1 laptop. For more experimental details, please refer to Appendix E.

**Experiment 1.** The distribution for eigenvalues of the matrix $\mathbf{A}$. In the error analysis (Theorem 1 and Theorem 2), the error is controlled by the approximation of $\mu$ to unobserved eigenvalues $\{\lambda_i\}_{i=m+1}^n$, i.e., $\max_{m+1 \le i \le n} |\lambda_i - \mu|$ and $\max_{m+1 \le i \le n}(\lambda_i - \mu)^2$. It further implies that the distribution of eigenvalues $\{\lambda_i\}_{i=m+1}^n$ is essential for the proposed method. Intuitively, if eigenvalues $\{\lambda_i\}_{i=m+1}^n$ cluster around a small interval, then the rough estimate (7) for $\mu$ is enough to approximate the unknown eigenvalues well. Conversely, if eigenvalues $\{\lambda_i\}_{i=m+1}^n$ spread across a large interval, then we cannot expect a single $\mu$ to estimate all eigenvalues $\{\lambda_i\}_{i=m+1}^n$. Here, we have four specially designed cases for distributions of eigenvalues of the matrix $\mathbf{A}$ to illustrate our theoretical observations for the proposed method. Case 1 (evenly spaced): all eigenvalues $\{\lambda_i\}_{i=1}^n$ are evenly spaced in $[-1, 1]$; Case 2 (separated): half of eigenvalues are far away from the remaining, i.e., eigenvalues $\{\lambda_i\}_{i=1}^{n/2}$ are evenly spaced in $[-1, -4/5]$ and the remaining eigenvalues $\{\lambda_i\}_{i=n/2+1}^n$ are evenly spaced in $[4/5, 1]$; Case 3 (right centered): the minimal $2\%$ of eigenvalues and the remaining $98\%$ of eigenvalues gather together respectively, i.e., eigenvalues $\{\lambda_i\}_{i=1}^{n/50}$ and $\{\lambda_i\}_{i=n/50+1}^n$ are spaced evenly in $[-1, 4/5]$ and $[4/5, 1]$ respectively; Case 4 (left centered): the maximal $2\%$ of eigenvalues and the remaining $98\%$ of eigenvalues gather together respectively, i.e., eigenvalues $\{\lambda_i\}_{i=1}^{49/50n}$ and $\{\lambda_i\}_{i=49/50n+1}^n$ are evenly spaced in $[-1, 4/5]$ and $[4/5, 1]$ respectively. The vector $\mathbf{b}$ is proportional to $[1, \cdots, 1]^{\mathrm{T}}$ with $\|\mathbf{b}\| = 0.1$. The remaining parameters are $n = 5 \times 10^3$ and $\rho = 0.1$. Here we adopt the first-order ASEM (i.e., $\mu$ is defined in (7)). Figure 1 validates our theories that the proposed algorithm converges fast if unknown eigenvalues $\{\lambda_i\}_{i=m+1}^n$ are close. Moreover, without the need to compute all eigenvalues, partial eigen information is enough to achieve satisfactory solutions in practice, except for the hard case (e.g., Case 4).

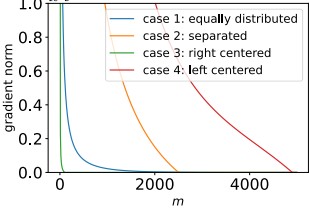 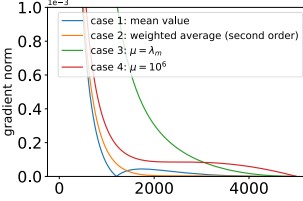 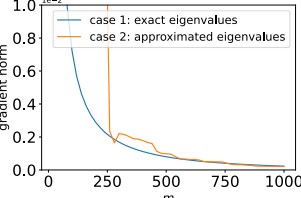

Figure 1: Trajectories of suboptimality (gradient norm $\|\nabla f_{\mathbf{A},\mathbf{b},\rho}(\mathbf{x})\|$) with different distributions for eigenvalues in Experiment 1.

Figure 2: Trajectories of suboptimality (gradient norm $\|\nabla f_{\mathbf{A},\mathbf{b},\rho}(\mathbf{x})\|$) with different $\mu$ in Experiment 2.

Figure 3: Trajectories of suboptimality (gradient norm $\|\nabla f_{\mathbf{A},\mathbf{b},\rho}(\mathbf{x})\|$) with exact and approximated eigenvalues and eigenvectors in Experiment 3.

**Experiment 2.** The effect of the parameter $\mu$ (first-order and second-order ASEMs). For the first- and second-order ASEMs, we define $\mu$ according to (7) and (10) respectively. Here, we test the effect of

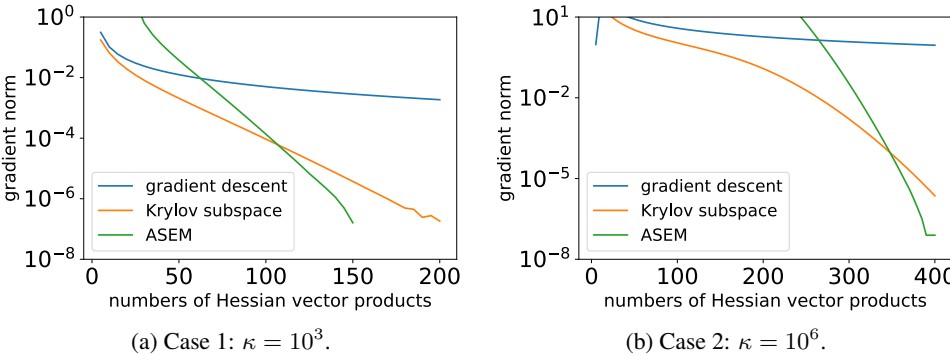

(a) Case 1: $\kappa = 10^3$.

(b) Case 2: $\kappa = 10^6$.

Figure 4: Trajectories of suboptimality (gradient norm $\|\nabla f_{\mathbf{A},\mathbf{b},\rho}(\mathbf{x})\|$) for ASEM, the Krylov subspace method and the gradient descent method in Experiment 4.

$\mu$ for the proposed method in solving the cubic regularized quadratic problem, with other parameters fixed. Case 1 (first-order ASEM): $\mu$ is adopted as the mean value of unknown eigenvalues, defined in (7); Case 2 (second-order ASEM): $\mu$ is selected as the weighted average of unknown eigenvalues with weights $c_i^2$ (see (10)); Case 3 (first-order ASEM): $\mu = \lambda_m$, the maximal eigenvalue we observe; Case 4: $\mu = 10^6$, much larger than the eigenvalues of $\mathbf{A}$, as an approximation to $+\infty$. The vector $\mathbf{b}$ is proportional to $[\lambda_1, \cdots, \lambda_n]^\mathrm{T}$ with length $\|\mathbf{b}\| = 0.1$. Eigenvalues of the matrix $\mathbf{A}$ are evenly spaced in $[-1, 1]$. The remaining parameters are $d = 5 \times 10^3$ and $\rho = 0.1$. Figure 2 shows the superiority of the second-order ASEM over the first-order ASEM that it is more stable and converges faster when $m$ is large, consistent with Theorem 1 and Theorem 2. Moreover, the results further imply the importance of the choice of $\mu$. There are several observations from Figure 2. Firstly, we cannot discard the residual term with the unknown eigenvalues $\{\lambda_i\}_{i=m+1}^n$, where they still contain much information, as is shown in Case 1 and Case 4. Secondly, the random selection of $\mu$ does not work well and may even cause divergence (e.g., see Case 3 and Case 4). Furthermore, a suitable choice of $\mu$ leads to a well-behaved algorithm (e.g., see Case 1 and Case 2).

**Experiment 3.** Approximation capabilities of the Krylov subspace method for ASEM. As is introduced in Section 4, we adopt $m$-dimensional Krylov subspace $\mathcal{K}_m(\mathbf{B}, \mathbf{u}_1)$ to approximately calculate $m$ algebraically smallest eigenvalues $\{\lambda_1, \cdots, \lambda_m\}$ and the corresponding eigenvectors $\{\mathbf{v}_1, \cdots, \mathbf{v}_m\}$. We now investigate the performance of ASEM with estimated eigenvalues and eigenvectors. The vector $\mathbf{b}$ is proportional to the vector $[1, \cdots, 1]^\mathrm{T}$ with length $\|\mathbf{b}\| = 0.1$. Eigenvalues of the matrix $\mathbf{A}$ are evenly spaced in $[-1, 1]$. The remaining parameters are $n = 5 \times 10^3$ and $\rho = 0.1$. We adopt the first-order ASEM (i.e., $\mu$ is defined in (7)) here. Trajectories for suboptimality with exact and approximated eigenvalues and eigenvectors are shown in Figure 3. The Krylov subspace $\mathcal{K}_m(\mathbf{B}, \mathbf{u}_1)$ with relatively low dimension $m$ for ASEM matches well with ASEM with exact eigenvalues. This experiment justifies the use of the $m$-dimensional Krylov subspace for $m$ eigenvalues and eigenvectors in ASEM.

**Experiment 4.** Comparison of ASEM with the Krylov subspace method [3, 2] and the gradient descent method [1] on synthetic problems. For large-scale problems, the Krylov subspace method and the gradient descent method are two state-of-the-art methods for CRS (1). In this experiment, we compare the proposed ASEM against the Krylov subspace method and the gradient descent method. The dominant computation steps for these three methods are Hessian-vector products ($\mathcal{O}(mn^2)$). The vector $\mathbf{b}$ is proportional to the vector $[1, \cdots, 1]^\mathrm{T}$ with length $\|\mathbf{b}\| = 0.1$. Eigenvalues of the matrix $\mathbf{A}$ are evenly spaced in $[-1, 1]$. Similar to the setting in [2], we define the condition number for (1) as $\kappa = \frac{\lambda_n + \rho \cdot \|\mathbf{x}^*\|}{\lambda_1 + \rho \cdot \|\mathbf{x}^*\|} = \frac{\lambda_n + \sigma^*}{\lambda_1 + \sigma^*}$. Then, we have $\sigma^* = \frac{\lambda_n - \kappa \cdot \lambda_1}{\kappa - 1}$ and $\rho = \frac{\sigma^*}{\|(\mathbf{A} + \sigma^* \mathbf{I})^{-1} \mathbf{b}\|}$. Case 1: easy case that $\kappa = 10^3$; Case 2: harder case that $\kappa = 10^6$. The remaining parameter is $n = 5 \times 10^3$. We adopt the first-order ASEM (i.e., $\mu$ is defined in (7)). As shown in Figure 4, ASEM outperforms both the gradient descent method and the Krylov subspace method when $m$ is relatively large. It is reasonable that ASEM underperforms when $m$ is small since the $m$-dimensional Krylov subspace cannot well approximate eigenvalues and eigenvectors of $\mathbf{A}$. The results further demonstrate the performance of the proposed ASEM method.

**Experiment 5.** Comparison of ASEM with the Cauchy point method [5], the gradient descent method and the Krylov subspace method on the CUTEst problems [7]. The CUTEst library collects various unconstrained and constrained optimization problems that arise in real applications. In this part, we compare the numerical performances of the ARC algorithm [3] on four unconstrained optimization problems from the CUTEst library, where subproblems are solved by the Cauchy point method (ARC-CP), the gradient descent method (ARC-GD), the Krylov subspace method (ARC-Krylov($k$), where $k$ is the number of Lanczos basis vectors) and the ASEM method (ARC-ASEM($m$), where $m$ is the number of eigenvalues for ASEM). The architecture of the ARC algorithm is provided in Appendix E.2. Four unconstrained optimization problems (e.g., TOINTGSS, BRYBAND, DIXMAANG, and TQUARTIC) in the CUTEst library are adopted for testing, where the dimensions are 1000, 2000, 3000, and 5000, respectively. We use the first-order ASEM ($\mu$ is defined in (7)) here since we found that the performances of the first-order ASEM and the second-order ASEM do not differ much for these problems. Numerical results are reported in Table 1, where $\mathbf{x}_{\text{out}}$, $\|\nabla f(\mathbf{x}_{\text{out}})\|$, $\lambda_1(\nabla^2 f(\mathbf{x}_{\text{out}}))$, iter and time represent the output of the ARC algorithm, the suboptimality (gradient norm), the minimal eigenvalue of the Hessian matrix, number of iterations for the ARC and CPU time, respectively. Here are several observations. Firstly, the proposed ASEM algorithm outperforms others in most cases and is comparable to the Krylov subspace method sometimes, where the ASEM achieves a worse suboptimality (gradient norms) or CPU time. Furthermore, only one eigenvalue is enough for the ASEM to perform well (i.e., ARC-ASEM($m$) with $m = 1$), which is surprising. For more experimental details, please refer to Appendix E.

Table 1: Results on CUTEst problems in Experiment 5.

| Problem | Method | $f(\mathbf{x}_{\text{out}})$ | $\|\nabla f(\mathbf{x}_{\text{out}})\|$ | $\lambda_1(\nabla^2 f(\mathbf{x}_{\text{out}}))$ | iter | time(s) |
|---|---|---|---|---|---|---|
| TOINTGSS ($n = 1000$) | ARC-CP | 3.60E+14 | 4.12E-05 | 1.40E-16 | 1000 | 6.08 |
| | ARC-GD | 3.60E+14 | 1.42E-06 | 3.89E-16 | 100 | 6.98 |
| | ARC-Krylov(1) | 3.60E+14 | 4.12E-05 | 1.20E-15 | 300 | 6.75 |
| | ARC-Krylov(10) | 3.60E+14 | 2.20E-08 | 1.29E-15 | 19 | **1.87** |
| | ARC-ASEM(1) | 3.60E+14 | **8.01E-10** | -1.63E-15 | 19 | 2.17 |
| | ARC-ASEM(10) | 3.60E+14 | 8.17E-10 | -7.67E-16 | 19 | 2.74 |
| BRYBAND ($n = 2000$) | ARC-CP | 7.49E+14 | 1.10E-03 | 5.40E+00 | 1000 | 8.27 |
| | ARC-GD | **1.25E+05** | 4.93E+03 | 4.40E+02 | 100 | 11.05 |
| | ARC-Krylov(10) | 7.49E+14 | 6.60E-06 | 5.40E+00 | 100 | 9.85 |
| | ARC-Krylov(30) | 7.49E+14 | 1.14E-07 | 5.40E+00 | 14 | 2.37 |
| | ARC-ASEM(1) | 7.49E+14 | 1.02E-07 | 5.40E+00 | 14 | **2.24** |
| | ARC-ASEM(10) | 7.49E+14 | **1.01E-07** | 5.40E+00 | 14 | 3.83 |
| DIXMAANG ($n = 3000$) | ARC-CP | 1.00E+00 | 3.13E-04 | 6.67E-04 | 2000 | 35.16 |
| | ARC-GD | 1.00E+00 | 9.24E-05 | 6.67E-04 | 200 | 33.83 |
| | ARC-Krylov(10) | 1.00E+00 | 3.44E-05 | 6.67E-04 | 500 | 32.18 |
| | ARC-Krylov(30) | 1.00E+00 | 9.06E-09 | 6.67E-04 | 46 | **6.65** |
| | ARC-ASEM(1) | 1.00E+00 | 5.53E-09 | 6.67E-04 | **30** | 7.51 |
| | ARC-ASEM(10) | 1.00E+00 | **4.85E-09** | 6.67E-04 | 42 | 18.74 |
| TQUARTIC ($n = 5000$) | ARC-CP | 8.04E-01 | 6.10E-02 | -5.41E-05 | 500 | 71.92 |
| | ARC-GD | 8.05E-01 | 2.77E-02 | -4.80E-05 | 100 | 98.14 |
| | ARC-Krylov(1) | 8.05E-01 | 2.76E-02 | -4.71E-05 | 100 | 29.43 |
| | ARC-Krylov(10) | **5.05E-14** | **8.48E-09** | 4.00E-04 | 46 | 16.09 |
| | ARC-ASEM(1) | 7.43E-14 | 9.62E-09 | 4.00E-04 | 46 | **15.47** |
| | ARC-ASEM(10) | 7.43E-14 | 9.62E-09 | 4.00E-04 | 46 | 16.18 |

# 6 Conclusion

We develop the first-order and the second-order truncated secular equations as surrogates to the secular equation with full eigendecomposition in solving the CRS (1). The proposed ASEM is an efficient alternative to existing methods for solving CRS since it reduces the computational cost from $\mathcal{O}(n^3)$ to $\mathcal{O}(mn^2)$. Our CRS solvers feature rigorous theoretical error bound, which is related to the amount of eigen information used. We also discuss in detail the implementation of our proposed algorithm ASEM. In particular, we show how only Hessian-vector products are needed, but not matrix inversion. Numerical experiments are conducted to further investigate the properties and performance

of the proposed ASEM and corroborate with the theoretical results. From our experiments, we find that the proposed ASEM is more efficient than the state-of-the-art methods on synthetic and CUTEst problems.

## Acknowledgement

We would like to thank the anonymous reviewers and chairs for their helpful comments. Michael K. Ng is supported by Hong Kong Research Grant Council GRF 12300218, 12300519, 17201020, 17300021, C1013-21GF, C7004-21GF and Joint NSFC-RGC N-HKU76921. Man-Chung Yue is supported by the Research Grants Council (RGC) of Hong Kong under the GRF project 15305321.

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
