# A  Proof of Theorem 1

The secular equation (4) can be decomposed into two parts:

$$w(\sigma) = \sum_{i=1}^{m} \frac{c_i^2}{(\lambda_i + \sigma)^2} + \sum_{i=m+1}^{n} \frac{c_i^2}{(\mu + \sigma)^2} - \frac{\sigma^2}{\rho^2} + \sum_{i=m+1}^{n} \frac{c_i^2}{(\lambda_i + \sigma)^2} - \sum_{i=m+1}^{n} \frac{c_i^2}{(\mu + \sigma)^2} \quad (14)$$
$$=: w_1(\sigma; \mu) + e_1(\sigma; \mu),$$

where $e_1(\sigma; \mu) = w(\sigma) - w_1(\sigma; \mu)$ represents the error between two secular equations.

By Taylor's theorem, we have

$$0 = w(\sigma^*) = w(\sigma_1^*) + \frac{d}{d\sigma} w(\xi) \cdot (\sigma^* - \sigma_1^*) = w_1(\sigma_1^*; \mu) + e_1(\sigma_1^*; \mu) + \frac{d}{d\sigma} w(\xi) \cdot (\sigma^* - \sigma_1^*),$$

where $\xi \in (\sigma^*, \sigma_1^*)$. By the definition that $w_1(\sigma_1^*; \mu) = 0$ and $w(\sigma)$ is monotonically decreasing in the domain (i.e., $\frac{d}{d\sigma} w(\xi) \neq 0$), we have

$$|\sigma_1^* - \sigma^*| \leq \frac{|e_1(\sigma_1^*; \mu)|}{\left| \frac{d}{d\sigma} w(\xi) \right|}. \quad (15)$$

For the error term, we can further rewrite it as

$$e_1(\sigma; \mu) = \sum_{i=m+1}^{n} \frac{c_i^2}{(\lambda_i + \sigma)^2} - \sum_{i=m+1}^{n} \frac{c_i^2}{(\mu + \sigma)^2} = \sum_{i=m+1}^{n} \frac{-2c_i^2}{(\varsigma_i + \sigma)^3} \cdot (\lambda_i - \mu),$$

for some $\varsigma_i \in (\mu, \lambda_i)$, $i = m+1, \cdots, n$. Due to $\mu \geq \lambda_m$ and $\lambda_i \geq \lambda_m$, we have $\varsigma_i > \lambda_m$ and thus $\varsigma_i + \sigma > \lambda_m - \lambda_1 > 0$. Therefore, the error term can be bounded by

$$|e_1(\sigma_1^*; \mu)| \leq \frac{2\|\mathbf{b}\|^2}{(\lambda_m - \lambda_1)^3} \cdot \max_{m+1 \leq i \leq n} |\lambda_i - \mu|. \quad (16)$$

The derivative of the secular equation is

$$\frac{d}{d\sigma} w(\sigma) = \sum_{i=1}^{n} \frac{-2c_i^2}{(\lambda_i + \sigma)^3} - \frac{2}{\rho^2}\sigma < 0,$$

for $\sigma > \max\{-\lambda_1, 0\}$. Then, we have

$$\left| \frac{d}{d\sigma} w(\xi) \right| = \sum_{i=1}^{n} \frac{2c_i^2}{(\lambda_i + \xi)^3} + \frac{2}{\rho^2}\xi \geq \sum_{i=1}^{n} \frac{2c_i^2}{(\lambda_i + \xi)^3} \geq \frac{2\|\mathbf{b}\|^2}{\max_{1 \leq i \leq n}(\lambda_i + \xi)^3} = \frac{2\|\mathbf{b}\|^2}{(\lambda_n + \xi)^3},$$

and

$$\left| \frac{d}{d\sigma} w(\xi) \right| = \sum_{i=1}^{n} \frac{2c_i^2}{(\lambda_i + \xi)^3} + \frac{2}{\rho^2}\xi \geq \frac{2}{\rho^2}\xi.$$

Moreover, $\sigma_1^*$ is a root of $w_1(\sigma; \mu) = 0$, which implies that

$$\sigma_1^{*2} = \rho^2 \cdot \left( \sum_{i=1}^{m} \frac{c_i^2}{(\lambda_i + \sigma_1^*)^2} + \sum_{i=m+1}^{n} \frac{c_i^2}{(\mu + \sigma_1^*)^2} \right) \leq \rho^2 \cdot \|\mathbf{b}\|^2 \cdot \frac{1}{(\lambda_1 + \sigma_1^*)^2},$$

and thus

$$0 \leq \sigma_1^* \leq \frac{-\lambda_1 + \sqrt{\lambda_1^2 + 4\rho \cdot \|\mathbf{b}\|}}{2} =: B_1.$$

Similarly, $\sigma^*$ is a root of $w(\sigma) = 0$, which further implies that

$$\sigma^{*2} = \rho^2 \cdot \sum_{i=1}^{n} \frac{c_i^2}{(\lambda_i + \sigma^*)^2} \leq \rho^2 \cdot \|\mathbf{b}\|^2 \cdot \frac{1}{(\lambda_1 + \sigma^*)^2},$$

and thus $0 \leq \sigma^* \leq B_1$. Due to $\xi \in (\sigma^*, \sigma_1^*)$ (or $\xi \in (\sigma_1^*, \sigma^*)$), we have

$$0 \leq \xi \leq B_1. \quad (17)$$

Equipped with the above results, we can bound the derivative term by

$$\left| \frac{d}{d\sigma} w(\xi) \right| \geq \max \left\{ \frac{2\|\mathbf{b}\|^2}{(\lambda_n + B_1)^3}, \frac{2B_1}{\rho^2} \right\}. \tag{18}$$

Note that $B_1 > -\lambda_1$ and thus $(\lambda_n + B_1)^3 > 0$, which implies that (18) makes sense. Combining with (15), (16) and (18), we have

$$|\sigma_1^* - \sigma^*| \leq \frac{2\|\mathbf{b}\|^2}{(\lambda_m - \lambda_1)^3} \cdot \min \left\{ \frac{(\lambda_n + B_1)^3}{2\|\mathbf{b}\|^2}, \frac{\rho^2}{2B_1} \right\} \cdot \max_{m+1 \leq i \leq n} |\lambda_i - \mu| =: C_m \cdot \max_{m+1 \leq i \leq n} |\lambda_i - \mu|,$$

where the constant $C_m$ is bounded by $\frac{2\|\mathbf{b}\|^2}{(\lambda_m - \lambda_1)^3} \cdot \min \left\{ \frac{(\lambda_n + B_1)^3}{2\|\mathbf{b}\|^2}, \frac{\rho^2}{2B_1} \right\}$.

## B   Proof of Theorem 2

The proof proceeds similarly to that of Theorem 1. The secular equation (4) can be decomposed into two parts:

$$
\begin{aligned}
w(\sigma) = & \sum_{i=1}^m \frac{c_i^2}{(\lambda_i + \sigma)^2} + \sum_{i=m+1}^n \frac{c_i^2}{(\mu + \sigma)^2} - 2 \sum_{i=m+1}^n \frac{c_i^2 \cdot (\lambda_i - \mu)}{(\mu + \sigma)^3} - \frac{\sigma^2}{\rho^2} \\
& + \sum_{i=m+1}^n \frac{c_i^2}{(\lambda_i + \sigma)^2} - \sum_{i=m+1}^n \frac{c_i^2}{(\mu + \sigma)^2} + 2 \sum_{i=m+1}^n \frac{c_i^2 \cdot (\lambda_i - \mu)}{(\mu + \sigma)^3} \\
=: & \ w_2(\sigma; \mu) + e_2(\sigma; \mu),
\end{aligned}
\tag{19}
$$

where $e_2(\sigma; \mu) = w(\sigma) - w_2(\sigma; \mu)$ represents the error between two secular equations. By Taylor's Theorem, we have

$$0 = w(\sigma^*) = w(\sigma_2^*) + \frac{d}{d\sigma} w(\xi) \cdot (\sigma^* - \sigma_2^*) = w_1(\sigma_2^*; \mu) + e_1(\sigma_2^*; \mu) + \frac{d}{d\sigma} w(\xi) \cdot (\sigma^* - \sigma_2^*),$$

where $\xi \in (\sigma^*, \sigma_2^*)$. By the definition that $w_2(\sigma_2^*; \mu) = 0$ and $w(\sigma)$ is monotonically decreasing in the domain (i.e., $\frac{d}{d\sigma} w(\varepsilon) \neq 0$), we have

$$|\sigma_2^* - \sigma^*| \leq \frac{|e_2(\sigma_2^*; \mu)|}{\left| \frac{d}{d\sigma} w(\xi) \right|}. \tag{20}$$

For the error term

$$e_2(\sigma; \mu) = \sum_{i=m+1}^n \frac{c_i^2}{(\lambda_i + \sigma)^2} - \sum_{i=m+1}^n \frac{c_i^2}{(\mu + \sigma)^2} + 2 \sum_{i=m+1}^n \frac{c_i^2 \cdot (\lambda_i - \mu)}{(\mu + \sigma)^3} = \sum_{i=m+1}^n \frac{3c_i^2}{(\varsigma_i + \sigma)^4} \cdot (\lambda_i - \mu)^2$$

for some $\varsigma_i \in (\mu, \lambda_i)$, $i = m + 1, \cdots, n$. Due to $\mu > \lambda_m$, we have $\varsigma_i > \lambda_m$ and thus $\varsigma_i + \sigma > \lambda_m - \lambda_1 > 0$. Therefore, the error term can be bounded by

$$|e_2(\sigma_2^*; \mu)| \leq \frac{3\|\mathbf{b}\|^2}{(\lambda_m - \lambda_1)^4} \cdot \max_{m+1 \leq i \leq n} (\lambda_i - \mu)^2. \tag{21}$$

Due to the special choice of $\mu = \mu_2$ defined in (10), same upper bound for the derivative term is achieved as that in Theorem 1, i.e.,

$$\left| \frac{d}{d\sigma} w(\xi) \right| \geq \max \left\{ \frac{2\|\mathbf{b}\|^2}{(\lambda_n + B_1)^3}, \frac{2B_1}{\rho^2} \right\}. \tag{22}$$

Combining with (20), (21) and (22), we have

$$|\sigma_2^* - \sigma^*| \leq \frac{3\|\mathbf{b}\|^2}{(\lambda_m - \lambda_1)^4} \cdot \min \left\{ \frac{(\lambda_d + B_1)^3}{2\|\mathbf{b}\|^2}, \frac{\rho^2}{2B_1} \right\} \cdot \max_{m+1 \leq i \leq n} (\lambda_i - \mu)^2 =: C_m \cdot \max_{m+1 \leq i \leq n} (\lambda_i - \mu)^2.$$

## C  Random Gaussian Matrix

In this part, we further elaborate on the error analysis in Theorem 1 for random Gaussian matrix, where we can access the eigen distribution. We first introduce the definition for a class of Gaussian random matrix.

**Definition 1.** *A $n \times n$ matrix $\mathbf{M}$ with entries $m_{ij}$ is a Wigner matrix if $m_{ij} = m_{ji}$ and $m_{ij}$ are randomly i.i.d. up to symmetry.*

**Definition 2.** *Let $\mathbf{M}$ be a real Wigner matrix with diagonal entries $m_{ii} \sim \mathcal{N}(0,1)$ and $m_{ij} \sim \mathcal{N}(0,1/2)$, $i \neq j$. Then, $\mathbf{M}$ is said to belong to the Gaussian orthogonal ensemble (GOE).*

**Remark.** Consider a random $n \times n$ matrix $\mathbf{W}$ with all entries coming from $\mathcal{N}(0,1)$, then $\mathbf{M} = \frac{\mathbf{W}+\mathbf{W}^{\mathrm{T}}}{2}$ is a Gaussian orthogonal ensemble (GOE) matrix.

**Theorem 3.** *Suppose that the $n \times n$ matrix $\mathbf{A} = \frac{\widetilde{\mathbf{A}}}{\sqrt{2n}}$ with $\widetilde{\mathbf{A}}$ belonging to GOE, then with probability of $1 - o(1)$, we have*

$$\max_{m+1 \leq i \leq n} |\lambda_i - \mu| \leq \mathcal{O}\left(\left(1 - \frac{m+1}{n}\right)^{2/3}\right), \tag{23}$$

*where $\mu \in [\lambda_{m+1}, \lambda_n]$ and $n$ is large enough.*

*Proof.* According to Section 1.4 in [6], eigenvalues $\{\lambda_1, \cdots, \lambda_n\}$ of $\mathbf{A}$ follows the Wigner's semi-circle law that p.d.f. of $\lambda \in \{\lambda_1, \cdots, \lambda_n\}$ satisfies

$$\lim_{n \to \infty} f_n(\lambda) = f(\lambda) = \frac{2}{\pi} \cdot \sqrt{1 - \lambda^2} \cdot \mathbb{I}_{[-1,1]}(\lambda),$$

where $\mathbb{I}_{[-1,1]}(\lambda)$ is the indicator function that values 1 if $\lambda \in [-1, 1]$ and 0 if $\lambda \notin [-1, 1]$. Therefore,

$$\begin{aligned} P(\lambda^* \leq \lambda \leq \lambda_{m+1}) &= P(\lambda \leq \lambda_{m+1}) - P(\lambda \geq \lambda^*) \\ &\approx \frac{m+1}{n} - \int_{-1}^{\lambda^*} f(s)\ ds \\ &= 1 - \frac{2}{\pi} \int_{-1}^{\lambda^*} \sqrt{1 - s^2}\ ds - \left(1 - \frac{m+1}{n}\right). \end{aligned}$$

Furthermore,

$$\begin{aligned} 1 - \frac{2}{\pi} \int_{-1}^{\lambda^*} \sqrt{1 - s^2}\ ds &= \frac{1}{\pi} \cdot \left(\frac{\pi}{2} - \arcsin(\lambda^*) - \lambda^* \cos(\arcsin(\lambda^*))\right) \\ &\approx \frac{4\sqrt{2}}{3\pi} \cdot (1 - \lambda^*)^{3/2} \\ &= \mathcal{O}\left((1 - \lambda^*)^{3/2}\right), \end{aligned}$$

where the last equation comes from $\frac{\pi}{2} - \arcsin(\lambda^*) \approx \sqrt{2(1 - \lambda^*)} = \mathcal{O}\left(\sqrt{1 - \lambda^*}\right)$ when $\lambda^* \to 1^-$, and we may select $\lambda^*$ with $1 - \lambda^* = \mathcal{O}\left(\left(1 - \frac{m+1}{n}\right)^{2/3}\right) \approx \left(\frac{3\pi}{4\sqrt{2}}\right)^{2/3} \cdot \left(1 - \frac{m+1}{n}\right)^{2/3}$ such that $P(\lambda^* \leq \lambda \leq \lambda_{m+1}) > 0$. Therefore, with probability of $1 - o(1)$, we have $\lambda^* \leq \lambda_{m+1} \leq \mu \leq \lambda_n \leq 1$ and thus

$$\max_{m+1 \leq i \leq n} |\lambda_i - \mu| \leq \lambda_n - \lambda_{m+1} \leq 1 - \lambda^* = \mathcal{O}\left(\left(1 - \frac{m+1}{n}\right)^{2/3}\right) \approx \left(\frac{3\pi}{4\sqrt{2}}\right)^{2/3} \cdot \left(1 - \frac{m+1}{n}\right)^{2/3}.$$

$\square$

**Remark.** The above theorem also holds for other kinds of symmetric random Gaussian matrix, e.g., the Gaussian unitary ensemble (GUE) matrix and the Gaussian symplectic ensemble (GSE) matrix etc., because eigen distribution for both GUE and GSE matrix follows the Wigner's semicircle law [6].

# D    Reduction to Diagonal Hessian Matrix

For the cubic regularized non-convex quadratic problem

$$\min_{\mathbf{x}} f_{\mathbf{A},\mathbf{b},\rho}(\mathbf{x}) := \mathbf{b}^{\mathrm{T}}\mathbf{x} + \frac{1}{2}\mathbf{x}^{\mathrm{T}}\mathbf{A}\mathbf{x} + \frac{\rho}{3}\|\mathbf{x}\|^3, \tag{24}$$

the Hessian matrix $\mathbf{A}$ is diagonalizable, i.e., there exist a diagonal matrix $\mathbf{\Lambda} \in \mathbb{R}^{n \times n} = \mathrm{diag}(\lambda_1, \cdots, \lambda_n)$ and a unitary matrix $\mathbf{V} \in \mathbf{R}^{n \times n}$, such that

$$\mathbf{A} = \mathbf{V}\mathbf{\Lambda}\mathbf{V}^{T}.$$

Then (24) is equivalent to the following problem (25) with diagonal Hessian matrix $\mathbf{\Lambda}$:

$$\min_{\mathbf{y}} f_{\mathbf{\Lambda},\mathbf{c},\rho}(\mathbf{x}) := \mathbf{c}^{\mathrm{T}}\mathbf{y} + \frac{1}{2}\mathbf{y}^{\mathrm{T}}\mathbf{\Lambda}\mathbf{y} + \frac{\rho}{3}\|\mathbf{y}\|^3, \tag{25}$$

where $\mathbf{c} = \mathbf{V}^{\mathrm{T}}\mathbf{b}$. Therefore, we adopt diagonal matrix $\mathbf{A}$ in numerical experiments since these two problems are equivalent.

# E    Further Details of Experiments

## E.1    The Cauchy Point

The Cauchy radius [5] for the cubic regularized quadratic problem

$$\min_{\mathbf{x}} f_{\mathbf{A},\mathbf{b},\rho}(\mathbf{x}) := \mathbf{b}^{\mathrm{T}}\mathbf{x} + \frac{1}{2}\mathbf{x}^{\mathrm{T}}\mathbf{A}\mathbf{x} + \frac{\rho}{3}\|\mathbf{x}\|^3$$

is the solution of

$$r = \arg\min_{\zeta \in \mathbb{R}} f_{\mathbf{A},\mathbf{b},\rho}(-\zeta \cdot \mathbf{b}/\|\mathbf{b}\|),$$

and the Cauchy point is defined as

$$\mathbf{x}_c = -r \cdot \mathbf{b}/\|\mathbf{b}\|.$$

Note that $f_{\mathbf{A},\mathbf{b},\rho}(\mathbf{x}_c) \leq f_{\mathbf{A},\mathbf{b},\rho}(\mathbf{0}) = 0$.

## E.2    The ARC Algorithm

We present the ARC algorithm [3] (Algorithm 1) here for completeness and clarification.

## E.3    Hyperparameters

In this subsection, we provide hyperparameters of the ARC algorithm in solving four optimization problems (e.g., TOINTGSS, BRYBAND, DIXMAANG and TQUARTIC) from the CUTEst library. In all experiments, we set $\gamma_1 = \gamma_2 = 2$, $\eta_1 = 0.1$, $\eta_2 = 0.9$ and $\rho_0 = 10^3$. The initial guess $\mathbf{x}_0$ is available for each problem (you can access it by *prob.x* command in MATLAB). For TOINTGSS, BRYBAND, DIXMAANG, and TQUARTIC, we select the top 1000, 2000, 3000, and 5000 variables as unknown variables for optimization, respectively.

## E.4    Additional Experiments

**Experiment 6.** The effect of the length $\|\mathbf{b}\|$. The derived upper bound in (6) and (11) are proportional to $\|\mathbf{b}\|^{3/2}$, which implies that larger $\|\mathbf{b}\|$ may cause worse approximation for $\sigma^*$ and thus the unattainable optimality. Here, we tested the convergence of the proposed method for different length $\|\mathbf{b}\|$ with other parameters fixed. In case 1 to 6, we set $\|\mathbf{b}\| \in \{1, 0.5, 0.2, 0.1, 0.05, 0.01\}$. The vector $\mathbf{b}$ is proportional to the vector $[1, \cdots, 1]^{\mathrm{T}}$. Eigenvalues of the matrix $\mathbf{A}$ are evenly spaced in $[-1, 1]$. The remaining parameters are $n = 5 \times 10^3$ and $\rho = 0.1$. We adopt the first-order ASEM (i.e., $\mu$ is defined in (7)) here. As is shown in Figure 5, the proposed method is robust with respect to the length $\|\mathbf{b}\|$, while the gradient descent [1] is ill-conditioned if $\|\mathbf{b}\|$ is tiny. Consistent with the error analysis, the proposed method converges faster for smaller length $\|\mathbf{b}\|$.

---

**Algorithm 1** The ARC algorithm

---

**Target:** a local minimizer of the objective function $f(\mathbf{x})$, where $\mathbf{x} \in \mathbb{R}^n$.
**Input:** initial guess $\mathbf{x}_0$, $\gamma_2 \geq \gamma_1 > 1$, $1 > \eta_2 \geq \eta_1 > 0$, $\rho_0 > 0$ and the integer $T > 0$.
**Output:** the approximate solution $\mathbf{x}_{\text{out}} \in \mathbb{R}^n$.

1: **for** $t = 0, \cdots, T$ **do**
2:      load $\mathbf{b}_t = \nabla f(\mathbf{x}_t)$, $\mathbf{A}_t = \nabla^2 f(\mathbf{x}_t)$
3:      compute the Cauchy point $\mathbf{s}_c$ of $f_{\mathbf{A}_t, \mathbf{b}_t, \rho_t}(\mathbf{s}) := \mathbf{b}_t^{\mathsf{T}} \mathbf{s} + \frac{1}{2} \mathbf{s}^{\mathsf{T}} \mathbf{A}_t \mathbf{s} + \frac{\rho_t}{3} \|\mathbf{s}\|^3$
4:      solve $f_{\mathbf{A}_t, \mathbf{b}_t, \rho_t}(\mathbf{s})$ by the gradient descent, the Krylov subspace or the ASEM method, the approximate solution is denoted as $\mathbf{s}_t$
5:      **if** $f_{\mathbf{A}_t, \mathbf{b}_t, \rho_t}(\mathbf{s}_c) \leq f_{\mathbf{A}_t, \mathbf{b}_t, \rho_t}(\mathbf{s}_t)$ **then**
6:          $\mathbf{s}_t = \mathbf{s}_c$;
7:      **end if**
8:      compute

$$\kappa_t = \frac{f(\mathbf{x}_t) - f(\mathbf{x}_t + \mathbf{s}_t)}{-f_{\mathbf{A}_t, \mathbf{b}_t, \rho_t}(\mathbf{s}_t)};$$

9:      set

$$\mathbf{x}_{t+1} = \begin{cases} \mathbf{x}_t + \mathbf{s}_t & \text{if } \kappa_t \geq \eta_1 \\ \mathbf{x}_t & \text{otherwise} \end{cases}$$

10:     set

$$\rho_{t+1} \in \begin{cases} (0, \rho_t] & \text{if } \kappa_t > \eta_2 & \text{(very successful iteration)} \\ [\rho_t, \gamma_1 \rho_t] & \text{if } \eta_1 \leq \kappa_t \leq \eta_2 & \text{(successful iteration)} \\ [\gamma_1 \rho_t, \gamma_2 \rho_t] & \text{otherwise} & \text{(unsuccessful iteration)} \end{cases}$$

11: **end for**

---

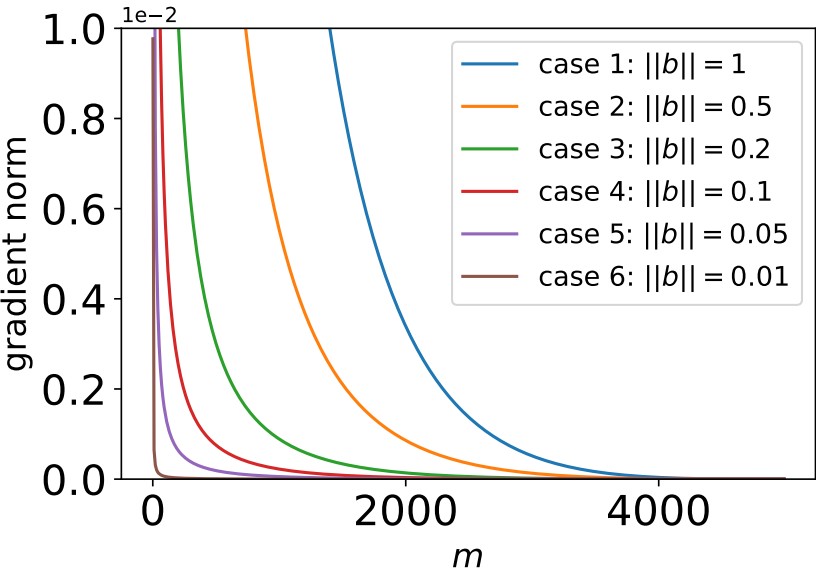

Figure 5: Trajectories of suboptimality (gradient norm $\|\nabla f_{\mathbf{A}, \mathbf{b}, \rho}(\mathbf{x})\|$) with different length $\|\mathbf{b}\|$ in Experiment 6.