# OpenReview forum: "Approximate Secular Equations for the Cubic Regularization Subproblem"
_NeurIPS.cc/2022/Conference — NeurIPS 2022 Accept_

### Official Review · Reviewer_QKHv · 2022-06-13

**Rating:** 4
**Confidence:** 4
**Soundness:** 3 good
**Presentation:** 2 fair
**Contribution:** 1 poor

**Summary:**

The well-known cubic regularization of Newton's method proposed by Nesterov (2006) requires solving a "cubic subproblem" at each iteration:
$
\min_{\mathbf{x} \in \mathbb{R}^{n}} =\mathbf{b}^{\mathrm{T}} \mathbf{x}+\frac{1}{2} \mathbf{x}^{\mathrm{T}} \mathbf{A} \mathbf{x}+\frac{\rho}{3}\|\mathbf{x}\|^{3}.
$
The cubic subproblem is equivalent to solving a "secular equation", which is a nonlinear equation that depends explicitly on the eigenvalues of $A \in \mathbb{R}^{n\times n}$. Solvers for this secular equation run in $O(n^3)$.

In this work the authors propose two "approximate" secular equations. Essentially the approximate secular equations are obtained by setting a parameter $m$ and replace all eigenvalues larger than $\lambda_m(A)$ in the secular equation with a constant $\mu\geq \lambda_m(A)$.  The advantage is that now we only need to compute the first $m$ eigenvalues of $A$, reducing the computational cost to $O(mn^2)$. The authors provide error analysis on the difference between the exact solution of the secular equation and the approximate solution. Furthermore, they provide numerical experiments to gauge the error of the approximate secular equation in a number of different scenarios. It is observed that the error depends on the distribution of the eigenvalues of $A$.

**Questions:**

The paper is clear and easy to understand. My only question is how does the approximate solver work in practice when used in cubic regularization? If the authors can implement their method and compared to state of art algorithms such as Carmon and Duchi 2018, or Cartis et al. 2011 and demonstrate a clear advantage, I would consider increasing my overall evaluation.

**Limitations:**

Yes. There is no direct negative societal impact.

**Strengths And Weaknesses:**

### Strengths
The ability to solve cubic subproblems efficiently is very important for the success of cubic regularization. The idea of solving an approximate cubic subproblem instead of an exact one can be useful in specific cases where the large eigenvalues of $A$ is concentrated in a small interval. The error bounds in this paper also become tighter in this regime.

### Weaknesses
- The main contribution of this paper is proposing two approximate secular equations and analyzing their errors. However, both the proposed equations and the error analysis are quite obvious and unsurprising. It basically boils down to the following idea: in the equation
$
w(\sigma)=\sum_{i=1}^{n} \frac{c_{i}^{2}}{\left(\lambda_{i}+\sigma\right)^{2}}-\frac{\sigma^{2}}{\rho^{2}} = 0,
$
we can pick an index $m$ and replace all eigenvalues greater or equal to $\lambda_m(A)$ with a constant $\mu$, resulting in an approximate equation. The larger $m$ is, the smaller the approximation error. When $m=n$, the method is exact. The error analysis is also based on this idea. The overall theoretical contribution is weak because the tradeoff between $m$ and the accuracy is quite obvious. It is also clear that the resulting error will depend on $m$, $\mu$ and the distribution of the eigenvalues. Using a small $m$, of course, will increase the error and reduce the running time, which is the gist of this paper.
- The motivation for this work is that the approximate secular equations can be used to solve the cubic subproblem, resulting in more efficient iterates for cubic regularization of Newton's method. However the experimental section does not contain such an experiment where an actual optimization problem is solved with cubic regularization and the running time is compared with other methods for solving the cubic subproblem, either exactly or approximately.
- In the end, there is neither a theoretical proof nor a convincing experimental section that demonstrate the advantage of the proposed method when it is actually used in cubic regularization.

---

> ### Author Response · Authors · 2022-07-28
> **Thanks so much for your comments. Here are our response**
>
> Thanks so much for your comments. **We uploaded the latest version according to your comments.**
>
> In [1], Carmon and Duchi used gradient descent to solve the subproblem (1) in the whole space $\mathbb{R}^{n}$; In [2] and [3], Cartis et al. approximately solved (1) in the Krylov subspace, which is a (much) lower dimensional space. In this paper, our main idea is to solve (1) by solving the approximate secular equations. **Our motivations and ideas are totally different from the state-of-the-art methods.**
>
> We have the following answers to your questions and comments.
>
> **Answers to your questions.**
>
> **(1)** We agree that we can intuitively imagine that larger $m$ implies more accuracy and there is a tradeoff between $m$ (computation costs) and accuracy. It is a natural and obvious phenomenon for most of approximation methods in mathematics, e.g., low-rank matrix decomposition, Krylov subspace method, and approximation capabilities of deep neural networks. In [10], they solve the secular equation and get accurate solution in each step of cubic regularization method. It performs well in low-dimensional problems. However, for high-dimensional problems, the eigendecomposition for large matrices is expensive. As is shown in Table 1 (the TQUARTIC problem), we solve the $5000$-dimensional problems in less than $16$ seconds, however, the full eigendecomposition for a dense $5000 \times 5000$ matrix may require more than $16$ seconds (but only one iteration). In this sense, the proposed ASEM makes sense.
>
> **(2)** Besides some numerical exploration on simply ASEM in solving (1), we also test the adaptive cubic regularization method (ARC), which is the most popular variant of the cubic regularization method, in solving some optimization problems in real applications. We put all details and results in **Experiment 5 and Table 1 in section 5**. **CUTEst is a library that collects various optimization problems from real applications**. For easier implementation, we don't need to know the background of application problems and directly use the CUTEst library to test the performance of optimization algorithms.
>
> **(3)** In the **Experiment 5**, we compare the proposed ARC-ASEM with ARC-CP (Cauchy point method), ARC-GD (gradient descent in **Carmon and Duchi 2019 [1]**) and ARC-Krylov (the Krylov subspace method in **Cartis et al. 2011 [3]**) in solving some optimization problems that arise in real applications (collected in CUTEst library). As is shown in Tabe 1, the proposed ARC-ASEM is significantly better than ARC-CP and ARC-GD but slightly better than ARC-Krylov. It is within our expectations as the Krylov subspace method is one of the most popular methods in computational math. However, the ARC-Krylov is sensitive to the number of subspaces but ARC-ASEM is more robust with the number of eigenvalues in real applications.

---

> > ### Comment · Reviewer_QKHv · 2022-08-07
> > **Response to authors**
> >
> > I thank the authors for their response. However I still believe that that my original point holds: the proposed method is very obvious and the theoretical results are weak. I would like to keep my current score.

---

> > > ### Author Response · Authors · 2022-08-08
> > > **Here are our response**
> > >
> > > Thanks for your response. We would like to clarify something further.
> > >
> > > **(1)** We didn't really agree the method is obvious and it is the first work in solving cubic subproblems by ASEM. Moreover, 'obvious' does not mean a lack of contribution. We suggest solving an approximate secular equation (ASEM) rather than the exact secular equation, where the former reduces much computation than the latter in large-scale problems. Our theoretical analysis is based on the approximate secular equation and shows that more eigen information has lower error. It is within our expectations. We found from the error bound that the error depends on the **'distribution' of eigenvalues, which is interesting**.  **To make it more clear, we extend the result for random Gaussian matrices, where we know the eigen distribution (please see Line 124-128, Page 4, equation (8) and full proofs are in the appendix).** Those observations are also validated in the numerical expenriments.  To the best of our knowledge, it is the first work that solves the cubic subproblem by ASEM.
> > >
> > > **(2)** You mentioned that your main question is how the proposed ASEM works in real applications, compared with state-of-the-art methods and you would consider increasing evaluation. We compared the ASEM with sota methods in real application problems. Please see Experiment 5, Table 1 in section 5. Are you satisfied with this part? Thanks.
> > >
> > > We are looking forward to your further comments.

---

> > > > ### Comment · Reviewer_QKHv · 2022-08-08
> > > > **Response to authors**
> > > >
> > > > Thank you again for the rebuttal. I have looked at the experimental section more carefully. Indeed it is a bit surprising to me that ASEM actually performs well even with one eigenvalue. I think this may perhaps be an interesting future direction that is worth looking into.
> > > >
> > > > I maintain my initial opinion that the theoretical contributions are still a bit weak. The fact that the final error of the approximate secular equation depends on the distribution of the eigenvalues are still quite obvious IMO. However, based on the experiments I have raised my score by 1.

---

> > > > > ### Author Response · Authors · 2022-08-08
> > > > > **Response to the reviewer**
> > > > >
> > > > > Thanks again for your comments. We really understand your point.
> > > > >
> > > > > We would like to state more about our contributions and confidence. **Our paper not purely theoretical...** The proposed algorithm is novel in solving cubic subproblems.
> > > > >
> > > > > ## Theory
> > > > > Numerically, gradient descent (Carmon and Duchi, 2019 [1]) is very slow... The Krylov subspace method is very a good method in real applications. The ASEM sometimes is comparable to or only slightly better than the Krylov subspace method. However, in [2], they analyze the convergence of the Krylov method, whose convergence is linear and sublinear. **However, the exact error bound should also vanish when $m=n$ for the Krylov subspace method, which is not observed in the error bounds of [2].**  In the analysis of ASEM, we have such observation from the error bound, which is more appropriate.
> > > > >
> > > > > ## Contribution
> > > > > In [2], their main contribution is the theoretical analysis of the Krylov subspace method. However, in this paper, **our main contribution** is the proposed novel ASEM in solving cubic subproblems, theoretical analysis of ASEM (and also on gaussian random matrices), numerical validation for the observation from theory and experimental test of the performances of the proposed method on some real applications. **It is not a purely theoretical paper...**
> > > > >
> > > > >
> > > > > ## Confidence
> > > > > We agree that [1] and [2] provide a very strong theoretical analysis for gradient descent and the Krylov subspace method. **We strongly believe the proposed method is useful in applications, providing another choice besides the Krylov-subspace method (which is the most popular in solving cubic subproblems).**

---

### Official Review · Reviewer_ZJXM · 2022-06-29

**Rating:** 7
**Confidence:** 4
**Soundness:** 4 excellent
**Presentation:** 4 excellent
**Contribution:** 3 good

**Summary:**

This paper proposes an efficient and general scheme for finding approximate solutions to the cubic regularization subproblem (CRS) used in the classic cubic regualization method in nonconvex optimization. The main advantage of the scheme lies in the efficient computation of the unique root in a given truncated approximate secular equation (ABE) that scales as $O(mn^2)$ where $n$ is the dimension of the underlying problem and $m < n$ is a parameter that balances the accuracy of the scheme with the overall computational cost. Numerical experiments are given to demonstrate the behavior of the scheme on different parameter choices, different kinds of problem instances, and a subset of the well-known CUTEst optimization problem dataset.

**Questions:**

1. Line 36: Do you mean $V\Lambda V^T$?

2. Proposition 2: Make this slightly more self-contained, by re-iterating what $v_1$ and $x^*$ are.

3. End of Section 1: A topic that might arouse more interest in the paper early on is a discussion on the choice of $m$ (which is discussed later, starting on line 211). Hence, it would be helpful to add 1-2 sentences at the end of Section 1 to say that this is a topic that will be discussed later in the paper.

4. The choice of $\mu$ in Theorem 2 does not seem to be immediately implementable, since it relies on $\lambda_{m+1},\ldots,\lambda_{n}$. However, equation (13) later on gives a tractable form for $\mu$. Perhaps a remark in (or after) the Theorem should be made to direct the reader to (13).

5. Line 197: "...if with imperfect initialization." Missing word?

6. Line 183: Return <- Returning

7. Line 289: "The rest of the parameter..." <- "The remaining parameter..."

8. The words "Krylov" and "Krylob" are used interchangeably throughout. I would suggest picking only one of these spellings.

**Limitations:**

The authors have sufficiently addressed all limitations (assumptions) in this paper.

**Strengths And Weaknesses:**

**Disclaimer**: My review is limited to the material presented in the 9-page body of the paper, and does not consider the materials in the supplement.

*Strengths*

1. The numerical experiments are significant in their scope and quality. Specifically, they test the behavior of the proposed scheme (ASEM) with respect to its key parameter $\mu$ and the distribution of eigenvalues in the CR subproblem. I also appreciate the inclusion of other well-known algorithms in the benchmarks, such as gradient descent, the Cauchy-point method, and the Krylov subspace method.

2. The error bounds in Theorem 1 and 2 are highly appreciated, as they encapsulate the expected behavior of the scheme and do abuse asymptotic notation to hide any universal constants.

3. The complexity improvement from $O(n^3)$ to $O(mn^2)$ is impactful both from a theoretical and practical point-of-view.

4. The writing of the paper is both clear and concise. Moreover, the remarks following some of the more important results, e.g., Proposition 3, are both welcome and informative.

*(Minor) Weaknesses*

1. There are few places that could better with additional clarifying statements (see the *Questions* section below).

2. There are few minor typos (see the *Questions* section below).

---

> ### Author Response · Authors · 2022-07-28
> **Thanks so much for your careful reviewing and comments. Here are our response.**
>
> Thanks so much and we really appreciate your careful review and constructive comments. **We uploaded the latest version according to your comments.**
>
> You mentioned our method is comparable with the Krylov subspace method. Yes, in the synthetic cases (Figure 4), our method is better but in real problems (CUTEst problems), ARC-ASEM is comparable (slightly better) with ARC-Krylov. It is within our expectations. We don't expect the ASEM significantly outperforms the Krylov subspace method as the Krylov subspace method is the most popular method in computational mathematics. But you can see in Table 1 that ASEM(1) and ASEM(10) are always good (is robust with $m$) but the Krylov subspace method is sensitive to the number of subspaces. In Table 1, the proposed ARC-ASEM(1) is comparable to or slightly better than ARC-Krylov.  In [1], Carmon and Duchi used gradient descent to solve the subproblem (1) in the whole space $\mathbb{R}^{n}$; In [2] and [3], Cartis et al. approximately solved (1) in the Krylov subspace, which is a (much) lower dimensional space. In this paper, our main idea is to solve (1) by solving the approximate secular equations. Our motivations and ideas are totally different from the state-of-the-art methods.
>
> We have the following answers to your questions and comments.
>
> **(1)** Sorry, it is a typo... We updated it in the current version colored in red.
>
> **(2)-(8)** Thanks for your comments, please see the updated version.

---

> > ### Comment · Reviewer_ZJXM · 2022-08-01
> > **Thank you + a small suggestion**
> >
> > Thank you for your hard work in adjusting the paper. After reading your changes, I only noticed that the red sentence in lines 77-78 is grammatically incorrect. The other material seems to be fine.
> >
> > Hence, I am maintaining my positive impression of your contribution.

---

> > > ### Author Response · Authors · 2022-08-03
> > > **Thanks so much for your pointing out.**
> > >
> > > Thanks so much! We revise the sentence accordingly. Please see the updated version.

---

### Official Review · Reviewer_CTtP · 2022-07-06

**Rating:** 5
**Confidence:** 3
**Soundness:** 3 good
**Presentation:** 3 good
**Contribution:** 2 fair

**Summary:**

The paper studies the cubic regularization technique and proposes two methods for approximating the secular equation based on the first-order and second-order Taylar expansions. Theoretical discussions including uniqueness and existence of the solution together with error analysis are provided. Five numerical experiments are conducted with comparable performance with the state of the art. However, the trade-off between accuracy improvement and computational reduction for these truncation-based approaches is not clear. Overall, the paper looks interesting with a variety of applications but with limited novelty.

**Questions:**

1. In line 212, what does the "computational waste" mean here?
2. In Table 1, it may not be fair to simply compare the overall running time. The running time per iteration could be compared as well. In addition, does the number of iterations depend on the selection of certain parameters in all methods being compared? If yes, please discuss which parameters are sensitive and provide guidelines for tuning if available.
3. In Section 5, it would be better to describe the computing platform and computer configurations at the very beginning of the section rather than in an unnoticeable place in Experiment 5.

**Limitations:**

Regularization techniques have been widely used in solving a lot of application problems, such as inverse problems and machine learning. But the paper does not show any such application experiment, which would limit the practical use and attraction.

**Strengths And Weaknesses:**

Strengths: The paper proposes to use simple Taylor approximations to approximate the secular equation, which intends to simplify the computation. Implementation details are provided, together with various numerical comparisons with the state of the art.

Weaknesses: The Taylor expansion-based truncation seems intuitive and standard, which will bring inaccuracy for the approximation. It is not fully clear how this type of approximation will affect the accuracy and reduce the computational cost. The general high-order approximation could be discussed as well. In addition, in the numerical results, say Table 1, ARC-Krylov seems to outperform the proposed method in terms of running time. A comparison of computational complexity in the big-O notation could be given with theoretical discussions.

---

> ### Author Response · Authors · 2022-07-28
> **Thanks so much for your comments. Here are our responses.**
>
> Thank you for your careful review and constructive comments.  **We uploaded the latest version according to your comments.** We have the following answers to your questions and comments.
>
>
> I would like to briefly introduce the cubic regularization method here. For a nonconstrained nonconvex optimization problem
> \begin{equation*}
>     \min_{\mathbf{x} \in \mathbb{R}^{n}} f(\mathbf{x}),
> \end{equation*}
> the main idea of the cubic regularization method is as follows. We first select an initialized point $x_0$. With the current state $x_{t}$, the next state $x_{t+1}$ satisfies the following cubic regularized problem:
>
> \begin{equation*}
>     x_{t+1} \in \arg \min_{\mathbf{x}} \left \langle  \nabla  f(x_{t}), x - x_{t} \right \rangle + \frac{1}{2} \left \langle x - x_{t}, \nabla^2 f(x_t) (x - x_{t})  \right \rangle + \frac{\rho_{t}}{3} ||x - x_{t}||_2^3.
> \end{equation*}
>
> The first two terms are Taylor expansion of $f(x)$ at $x_t$ up to the second order and the cubic regularization term constrains that $x_{t+1}$ cannot be far away from $x_t$ to guarantee the accuracy of the expansion. The ARC Algorithm adaptively updates $\rho_t$ based on the current state $x_{t}$. Please see Algorithm 1 (the ARC Algorithm) in the supplement or in [3]. In this paper, we aim to solve the above subproblem (cubically regularized quadratic problem) for the cubic regularization method.
>
> In [1], Carmon and Duchi used gradient descent to solve the subproblem (1) in the whole space $\mathbb{R}^{n}$; In [2] and [3], Cartis et al. approximately solved (1) in the Krylov subspace, which is a (much) lower dimensional space. In this paper, our main idea is to solve (1) by solving the approximate secular equations. Our motivations and ideas are totally different from the state-of-the-art methods.
>
> We agree the most **limitation** is that the trade-off between accuracy improvement and computational reduction is not clear. It is hard to mathematically and quantitatively show the trade-off since cubic regularization of Newton's method (or its most widely used variant ARC) requires iteratively solving the subproblem (1) multiple times. As is shown in Table 1 (the TQUARTIC problem), we solve the $5000$-dimensional problems in less than $16$ seconds, however, the full eigendecomposition for a $5000 \times 5000$ matrix may require more than $16$ seconds (but only one iteration). In this sense, the proposed ASEM makes sense. However, for low-dimensional problems (for example, $10$-d optimization problems), the full eigendecomposition is fast, so it is unnecessary to adopt ASEM or Krylov subspace method for ARC. As is shown in most of the previous works (e.g., [1] for gradient descent and [2,3] for the Krylov subspace method), more accuracy implies more costs. The trade-off between them depends on the optimization problem itself.  The approximation method performs well, especially in high-dimensional problems since full eigendecomposition is really expensive while calculating $m$ eigenvalues ($m << n$) is much cheaper.
>
>
> Yes, in synthetic cases, the proposed ASEM is better than Krylov (see Figure 4), but we don't expect the ASEM significantly outperforms the Krylov subspace method in solving real problems as the Krylov subspace method is the most popular method in computational mathematics. But you can see in Table 1 that ARC-ASEM(1) and ARC-ASEM(10) are always good (is robust with $m$) but the Krylov subspace method is sensitive to the number of subspaces. In Table 1, the proposed ARC-ASEM(1) is comparable to or slightly better than ARC-Krylov. In [3], the convergence of ARC is still guaranteed if we approximately solve the subproblem (1). Therefore, we may focus more on solving the subproblem. Our idea is novel and interesting.
>
> **Answers to your questions.**
>
> **(1)** If we accurately solve each subproblem of ARC (as in [10]), it could be very slow since full eigendecomposition is expensive for high-dimensional matrices. Similarly, if $m$ is large, that means the subproblem (1) is solved more accurately, then the total costs of ARC may be larger. As is shown in Table 1, ARC-ASEM(1) is better than ARC-ASEM(10) in terms of running time for convergence. Overly large $m$ for the proposed ASEM may cause 'waste' in total computation for convergence. Therefore, the selection of $m$ is important, which is related to the tradeoff between the computation and the accuracy. However, in real applications (shown in Table 1), we found that ASEM with $m=1$ is good enough.
>
> **(2)** The running time per iteration could be computed by (total running time)/(iterations), where both two are included in the table. For fairness, all hyperparameters of ARC are fixed (the same) for all experiments. Moreover, the most important parameter $\rho$ is adaptively updated in ARC. The only parameter is the selection of $m$ and we can see in Table 1 that ASEM is not sensitive with $m$ but the performances of the Krylov subspace method differ much on the number of subspaces.

---

> ### Author Response · Authors · 2022-07-28
> **Continue from the previous response**
>
> **Answers to your questions and comments**
>
> **(3)** Thanks for your suggestion, we move the information of the computing platform to the beginning of the section.
>
> **(4)** We tested the cubic regularization algorithm (ARC) with the proposed ASEM, Krylov subspace method, gradient descent and the Cauchy point method in solving real optimization problems (CUTEst) and all results are shown in **Table 1, Experiment 5**. **CUTEst is a library that collects various optimization problems from real applications**. For easier implementation, we don't need to know the background of application problems and directly use the CUTEst library to test the performance of optimization algorithms. Moreover, we also test the algorithm in solving high-dimensional real problems in CUTEst ($n=5000$).

---

> ### Author Response · Authors · 2022-08-08
> **Do you have further comments regarding our response. Thanks!**
>
> Dear reviewer CTtP,
>
> Thanks again for your review and comments. Do you have further comments? We hope our response answers all your concerns well.
>
> We would like to state something further.
> ## Contribution
> in this paper, **our main contribution** is the proposed novel ASEM in solving cubic subproblems, theoretical analysis of ASEM (and also on gaussian random matrices), numerical validation for the observation from theory and experimental test of the performances of the proposed method on some real applications. **It is not a purely theoretical paper...**
>
>
> ## Confidence
> **We strongly believe the proposed method is useful in applications, providing another choice besides the Krylov-subspace method (which is the most popular in solving cubic subproblems).**

---

### Official Review · Reviewer_MvBV · 2022-07-12

**Rating:** 5
**Confidence:** 1
**Soundness:** 3 good
**Presentation:** 3 good
**Contribution:** 2 fair

**Summary:**

In the cubic regularization method, a key step is to solve the so called secular equation which may costs O(n^3) time.
In this paper, the authors consider two faster approximation to the secular equation which reduce the computing time to O(n^2 m).
The authors give some theoretical analysis of the proposed methods.


**Questions:**

line 41: Proposition 2. Is this proposition adapted from Claim 2.1 in [1]? Why A + rho ||x^*|| > 0? Can you provide a proof of Proposition 2?
line 45: "and hence the gradient norm serves as an optimality measure" I do not understand this.
line 49: "lambda_1 + simga > 0" Should it be >= 0?
line 90: "for any mu". Should it be "for any mu > lambda_m"?
line 109: What does the symbol big O mean in your context? From line 113, it seems tht ||tuilde x - x*|| has no absolute upper bound, for example, suppose |lmabda_i| and simga* are very small.
line 122 -- 126: These discussions lack mathematical rigor.

From the theoretical view, is it possible to improve the bound?

**Limitations:**

Yes

**Strengths And Weaknesses:**

The proposed method reduces the computing time of secular equation, which may be novel.
The paper is relatively well-written and clear.
This paper is mostly a theoretical paper.
In my opinion, however, the theoretical results are not significant enough.
In fact, from the authors' theoretical results (Proposition 3 for example), the proposed method has an irreducible error.
From the authors' experiments, it may be the case that the obtained theoretical bound is too loose.

---

> ### Author Response · Authors · 2022-07-28
> **Thanks so much for your carefully reviewing. We would like to clarify something more clearly according to your comments**
>
> Thanks so much for your comments. **We uploaded the latest version according to your comments.** We have the following answers to your questions and comments.
> I would like to briefly introduce the cubic regularization method here. For a nonconstrained nonconvex optimization problem
> \begin{equation*}
>     \min_{\mathbf{x} \in \mathbb{R}^{n}} f(\mathbf{x}),
> \end{equation*}
> the main idea of the cubic regularization method is as follows. We first select an initialized point $x_0$. With the current state $x_{t}$, the next state $x_{t+1}$ satisfies the following cubic regularized problem:
>
> \begin{equation*}
>     x_{t+1} \in \arg \min_{\mathbf{x}} \left \langle  \nabla  f(x_{t}), x - x_{t} \right \rangle + \frac{1}{2} \left \langle x - x_{t}, \nabla^2 f(x_t) (x - x_{t})  \right \rangle + \frac{\rho_{t}}{3} ||x - x_{t}||_2^3.
> \end{equation*}
>
> The first two terms are Taylor expansion of $f(x)$ at $x_t$ up to the second order and the cubic regularization term constrains that $x_{t+1}$ cannot be far away from $x_t$ to guarantee the accuracy of the expansion. The ARC Algorithm adaptively updates $\rho_t$ based on the current state $x_{t}$. Please see Algorithm 1 (the ARC Algorithm) in the supplement or in [3]. In this paper, we aim to solve the above subproblem (cubically regularized quadratic problem) for the cubic regularization method.
>
> In [1], Carmon and Duchi used gradient descent to solve the subproblem (1) in the whole space $\mathbb{R}^{n}$; In [2] and [3], Cartis et al. approximately solved (1) in the Krylov subspace, which is a (much) lower dimensional space. In this paper, our main idea is to solve (1) by solving the approximate secular equations. Our motivations and ideas are totally different from the state-of-the-art methods.
>
> Our theoretical bound is different from some common bounds (e.g., linear convergence and quasi-linear convergence) since we derived it from solving the secular equation. For a $n \times n$ matrix, we can get an accurate solution (the error bound should be zero) if we set $m=n$. Moreover, we have an additional requirement that $m \leq n$, where the error bound should vanish at $m=n$ (it cannot be achieved for linear convergence). Therefore, the theoretical error bound should be different from some conventional bounds (e.g., linear convergence). In this way, our theoretical bound makes sense.
>
> **Line 41**. Yes, it is derived from Proposition 2.1 and 2.2 in [1]. I would like to give detailed proof here and will include it in the supplementary material. The global solution $x^{\star}$ of the cubic regularized problem (1) must satisfy (2) and (3) which are first-order and second-order conditions. We multiply both two sides of
> \begin{equation*}
>     (A + \rho ||x^{\star}|| \mathbf{I})x^{\star} + b = 0
> \end{equation*}
> by $\mathbf{v}_1$ (the eigenvector corresponds to the minimal eigenvalues of $\mathbf{A}$), then we have
> \begin{equation*}
>     (\lambda_1 + \rho ||x^{\star}|| ) \cdot (\mathbf{v}_1^{\mathrm{T}}x^{\star}) + \mathbf{v}_1^{\mathrm{T}} \mathbf{b} = 0.
> \end{equation*}
> If $\mathbf{v}_1^{\mathrm{T}} \mathbf{b} \neq 0$, then $\lambda_1 + \rho ||x^{\star}|| $ must be non-zero (thus strictly positive). Therefore, the matrix $\mathbf{A} + \rho ||x^{\star}|| \mathbf{I} \succ \mathbf{0}$ is positive definite and the solution $x^{\star} = -(\mathbf{A} + \rho ||x^{\star}|| \mathbf{I})^{-1} \mathbf{b}$. Moreover, if $\lambda_1 + \rho ||x^{\star}|| $ is positive, the solution of problem (1) is unique (please see Theorem 3.1 in [3]).
>
>
>
>
> **Line 45.** Under the condition that $v_1^{T} b \neq 0$, we know that the problem (1) has a unique solution, and more importantly, it is also a \textbf{unique stable point}. There does not exist another point $\bar{x} \neq x^{\star}$ such that $\nabla f_{\mathbf{A},\mathbf{b},\rho}(\bar{x}) = 0$. Without the loss of generality, we assume that $||\bar{x}||_2 > ||x^{\star}||$ (if $||\bar{x}||_2 = ||x^{\star}||$, the only solution for the first-order condition is $x^{\star} = -(\mathbf{A} + \rho ||x^{\star}|| \mathbf{I})^{-1} \mathbf{b}$. Then, we have $x^{\star} = -(\mathbf{A} + \rho ||x^{\star}\| \mathbf{I})^{-1} \mathbf{b}$ and $\bar{x} = -(\mathbf{A} + \rho ||\bar{x}|| \mathbf{I})^{-1} \mathbf{b}$.
>
> However,  $||(A + \rho ||\bar{x}||I)^{-1} b|| \leq  ||(A + \rho ||x^{\star}|| I)^{-1} b||$
> , which is a contradiction. Therefore, in numerical experiments, we use the gradient $||\nabla f_{\mathbf{A},\mathbf{b},\rho}(\mathbf{x}) = 0||_2$ to measure the optimality, where smaller gradient means to be closer to the global minima of the problem (1).

---

> ### Author Response · Authors · 2022-07-28
> **Continue from the previous response**
>
> **Line 90.** Yes, thanks for your pointing out the mistake here. It should be "$\mu \geq \lambda_{m}$".
>
> **Line 109 and 113.**
> Firstly, the solution $\sigma_1^{\star}$ of the approximate secular equation and the solution $\sigma^{\star}$ of the secular equation must satisfies that $\lambda_1 + \sigma_1^{\star} > 0$ and $\lambda_1 + \sigma^{\star} > 0$. Then, by the Taylor expansion, we have
> \begin{equation*}
>     (\lambda_i + \sigma_1^{\star})^{-1} = (\lambda_i + \sigma^{\star})^{-1} -  (\lambda_i + \sigma^{\star})^{-2} \cdot (\sigma_1^{\star} - \sigma^{\star}) + 2(\lambda_i + \sigma^{\star})^{-3} \cdot (\sigma_1^{\star} - \sigma^{\star})^2 + \cdots.
> \end{equation*}
> Note that $\lambda_1 + \sigma^{\star} > 0$ is fixed for the given $\mathbf{A}$, $\mathbf{b}$ and $\rho$ and is independent with the approximate solution $\sigma_1^{*}$ from the proposed method. When $\sigma_1^{\star} - \sigma^{\star}$ is small enough, then dominant term in the above expansion is the first-order term. Therefore, the big $\mathcal{O}(\cdot)$ here is consistent with that in the mathematical literature.
>
> **Line 122-126.** We put detailed mathematical proof in the supplement due to the page limit. All results are within theoretical guarantees. For the typical random Gaussian matrix, its eigenvalues follow the so-called semi-circle law with high probabilities. As we mentioned that the proposed ASEM depends on the distribution of eigenvalues of $\mathbf{A}$. Then we derived the explicit error bound for the ASEM for such cases.
>
> **Is it possible to improve the bound?** It is possible but may be hard. It is derived by solving approximate secular equations. The exact error bound should depend on the distribution of eigenvalues (as is shown in the numerical experiments, Figure 1). I think it is tight enough since the information delivered by the error bound is validated in the experiments (please see Figures 1, 2 and 3 in the experimental part). Also, our error bounds vanish at $m=n$, which is consistent with the exact error.

---

> ### Author Response · Authors · 2022-08-08
> **Thanks for your review. Do you have further comments?**
>
> Dear reviewer MvBV,
>
> Thanks again for your review and comments. Do you have further comments? We hope our response answers all your concerns well.
>
> We would like to state something further.
> ## Contribution
> in this paper, **our main contribution** is the proposed novel ASEM in solving cubic subproblems, theoretical analysis of ASEM (and also on gaussian random matrices), numerical validation for the observation from theory and experimental test of the performances of the proposed method on some real applications. **It is not a purely theoretical paper...**
>
>
> ## Confidence
> **We strongly believe the proposed method is useful in applications, providing another choice besides the Krylov-subspace method (which is the most popular in solving cubic subproblems).**

---

### Meta-Review · Area_Chair_YdLB · 2022-08-24

**Recommendation:** Accept
**Confidence:** Certain

**Metareview:**

This paper proposes a new method for solving the cubic subproblem in the cubic regularized Newton method. The propose method is simple, but works very well in practice. The numerical experiments demonstrated that the ARC algorithm combined with the proposed new subproblem solver significantly outperforms ARC with different subproblem solvers. Moreover, the accuracy of the solution generated by the proposed method can be several orders better than the one generated by other methods. Error analysis of the proposed method is also provided. Overall, this is a very nice contribution to the cubic regularized Newton method, which has the potential to accelerate this important method.

**Award:**

No

---

### Decision · Program_Chairs · 2022-09-14

Accept